# Finite-State Autoregressive Entropy Coding for Efficient Learned Lossless Compression

**Yufeng Zhang**[1,2]*, **Hang Yu**[2]†, **Jianguo Li**[2]†, **Weiyao Lin**[1]†
[1]Shanghai Jiao Tong University, [2]Ant Group

## Abstract

Learned lossless data compression has garnered significant attention recently due to its superior compression ratios compared to traditional compressors. However, the computational efficiency of these models jeopardizes their practicality. This paper proposes a novel system for improving the compression ratio while maintaining computational efficiency for learned lossless data compression. Our approach incorporates two essential innovations. First, we propose the Finite-State AutoRegressive (FSAR) entropy coder, an efficient autoregressive Markov model based entropy coder that utilizes a lookup table to expedite autoregressive entropy coding. Next, we present a Straight-Through Hardmax Quantization (STHQ) scheme to enhance the optimization of discrete latent space. Our experiments show that the proposed lossless compression method could improve the compression ratio by up to 6% compared to the baseline, with negligible extra computational time. Our work provides valuable insights into enhancing the computational efficiency of learned lossless data compression, which can have practical applications in various fields. Code is available at `https://github.com/alipay/Finite_State_Autoregressive_Entropy_Coding`.

## 1 Introduction

Lossless data compression has been a prominent area of interest in both academic research and industry. By reducing the amount of space required for data storage and minimizing the bandwidth necessary for data communication, lossless compression codecs hold great potential for applications in a broad range of computing and communication systems.

A practical lossless compression codec requires both superior compression ratios and computational efficiency, but most existing methods fail to achieve both simultaneously. Traditional methods (Boutell, 1997; webmproject, 2023) rely on hand-crafted codecs that **offer efficiency but yield suboptimal compression ratios**. Conversely, recent advances in machine learning have introduced generative models, including Autoencoders (Kingma & Welling, 2013; Burda et al., 2015), GANs (Mirza & Osindero, 2014; Arjovsky et al., 2017), Flow models (Dinh et al., 2016; Kingma & Dhariwal, 2018) and Autoregressive models (van den Oord et al., 2016b; Parmar et al., 2018), which exhibit remarkable potential in modeling data likelihood. These models can automatically learn codecs for different domains given sufficient training data. However, **while these models have demonstrated improved compression ratios compared to the traditional methods, they suffer from high time complexity**, rendering them impractical for general-purpose computation devices such as CPUs.

In this paper, we target at **improving the compression ratio** while **maintaining the computational efficiency** for lossless compression. Our focus lies on latent space models, specifically autoencoders, due to their ease of optimization and efficient computation (Mentzer et al., 2018; Townsend et al., 2019b; Ballé et al., 2018; Minnen et al., 2018). Most existing frameworks in this direction could be built upon the Asymmetric Numeral System (ANS) (Duda, 2009) for latent entropy coding, and can be categorized into three groups: continuous random latent space, discrete deterministic latent space, and discrete autoregressive latent space. The decompression workflows of these frameworks are illustrated in Figure 1a, Figure 1b and Figure 1c, respectively. The first group, based on continuous latent spaces (Townsend et al., 2019a; Kingma et al., 2019; Theis & Ho, 2021; Ruan et al., 2021), typically employs bits-back coding(Townsend et al., 2019a) (cf. Figure 1b). Although the continuous

---

*This work was done when Yufeng Zhang was a research intern at Ant Group.
†Corresponding Authors.

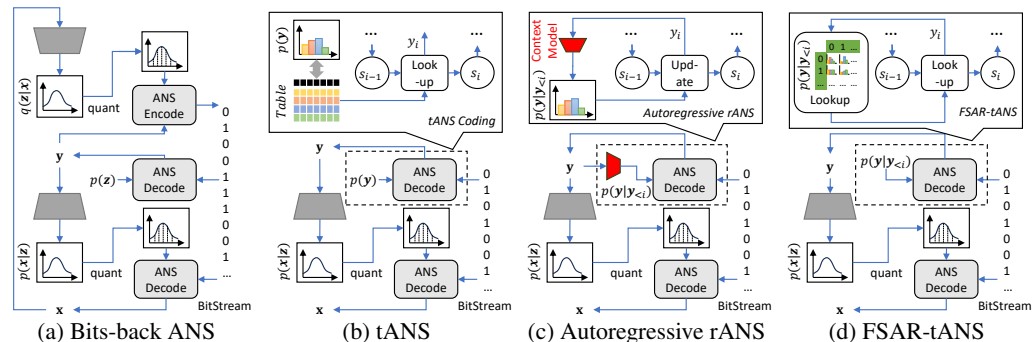

Figure 1: Different ANS-based decompression workflows for latent space models. (a) Bits-back ANS (Townsend et al., 2019a) for continuous random latent space. (b) table ANS (tANS) (Duda, 2013) coder for discrete deterministic latent space. (c) Autoregressive range ANS (rANS) (Duda, 2013) coder for discrete autoregressive latent space. (d) The proposed FSAR-tANS coder for discrete autoregressive latent space.

latent space facilitates optimization and achieves a favorable compression ratio, the decompression process becomes complex and slow due to the additional ANS encode step via the inference model. By contrast, frameworks (Ballé et al., 2018; Mentzer et al., 2018) utilizing discrete deterministic latent spaces enable fast decompression using a low-complexity table ANS (tANS) coder (Duda, 2013) (cf. Figure 1b). However, the representation power of independent discrete space is limited, and optimizing the discrete latent space is notoriously challenging, resulting in an inferior compression ratio. To mitigate this issue, the third group of works (Minnen et al., 2018; Lee et al., 2018; Cheng et al., 2020) improves the compression ratio by assuming flexible autoregressive dependence among the discrete latent variables instead of strict independence. However, similar to the continuous latent space, this improvement comes at the cost of decompression speed, as autoregressive models require a lengthy sequential decompression process that involves a complex context model in each iteration (cf. Figure 1c). Ultimately, existing frameworks fail to simultaneously achieve a satisfactory compression ratio and computational efficiency.

In contrast to the aforementioned works, our proposed method combines the best of both worlds. It features an efficiently coded discrete autoregressive latent space with a flexible latent model that can fit the input data well while robustly optimizing this latent space. We first develop an efficient autoregressive model called the **Finite-State AutoRegressive (FSAR)** model as the latent prior, which can be easily implemented with a lookup table, as illustrated in Figure 1d. Compared with other autoregressive models, the FSAR implementation has a similar time complexity to the low-complexity tANS, only adding an extra lookup step to the original tANS implementation. Thus, it achieves an excellent compression ratio with the help of autoregressive models while keeping the computation as efficient as tANS. Moreover, to better optimize the discrete latent space as required by the FSAR, we propose **Straight-Through Hardmax Quantization (STHQ)** based on the straight-through estimator (Bengio et al., 2013). STHQ further strengthens the representation ability of the latent space and improves the compression ratio. To summarize, our contributions are listed as follows:

- The Finite-State AutoRegressive (FSAR) entropy coder is proposed for flexible discrete latent space coding. It combines a low-complexity autoregressive Markov model with a fast entropy coder to achieve efficient latent coding.

- An optimization scheme called Straight-Through Hardmax Quantization (STHQ) is proposed for robust optimization of the discrete latent space. It enables gradient descent via the deterministic quantization process while maximizing the latent entropy for better likelihood estimation.

- Extensive experiments demonstrate that our proposed method improves the compression ratio by up to 6% compared to commonly used discrete deterministic autoencoders, with negligible additional computational time. Moreover, it is over 300 times faster than commonly used discrete autoregressive models while achieving a similar compression ratio.

## 2    RELATED WORKS

In this section, we review learned compression methods based on discrete deterministic latent space models and discrete autoregressive latent space models, as they are directly relevant to our work. For a brief survey on continuous random latent space models, we refer the readers to Appendix A.

**Discrete Deterministic Latent Space** Numerous compression methods employ discrete deterministic autoencoders due to their simplicity in latent coding. However, quantizing continuous latent space to discrete counterparts while enabling gradient calculation for optimization poses a challenge.

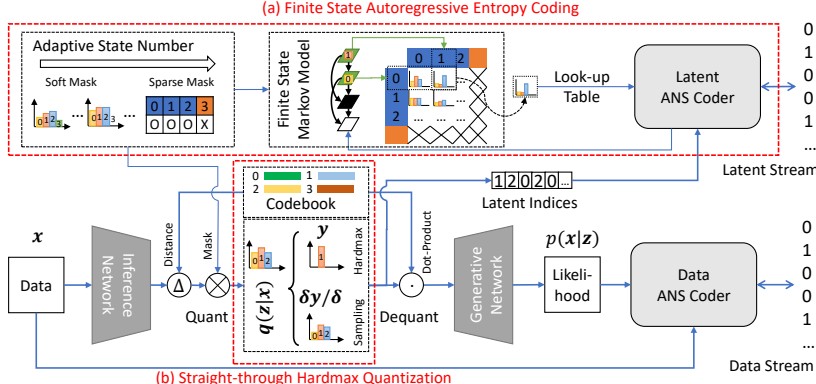

Figure 2: Overview of the proposed method. Here **x** represents the input data, **z**, **y** represents latent variables and samples respectively. ANS are employed for entropy coding of both the latent space and the data. Our major contributions are highlighted in red dashed boxes: (a) Finite-State Autoregressive Entropy Coding for latent entropy coding. (b) Straight-Through Hardmax Quantization for latent space optimization scheme.

To address this, universal quantization and soft scalar quantization have been applied respectively in (Ballé et al., 2018) and (Mentzer et al., 2018), allowing for gradient pass-through while simulating quantization. However, these methods overlook the complexity of the high-dimensional latent space, resulting in poor approximation. Alternatively, VQ-VAE (van den Oord et al., 2017) introduces vector quantization (VQ) and improves latent modeling through an additional codebook. Subsequent works, such as (Razavi et al., 2019; Esser et al., 2020), adopt this approach, employing trainable codebooks to enhance the quality of generative models. However, optimizing VQ-VAE is difficult as it requires an auxiliary loss function similar to the K-means algorithm. To counteract this problem, Sønderby (2017) proposed the continuous relaxation of VQ, representing the VQ process as sampling from a categorical distribution and enabling optimization using the ELBO similar to VAEs. This idea was further extended by utilizing hierarchical relaxed VQ-VAEs to model more complex latent spaces (Zhu et al., 2022). However, this relaxation breaks the deterministic nature of VQ. Instead, Agustsson et al. (2017) introduce soft VQ with deterministic annealing to enable optimization through hard quantization, but the practical implementation of the annealing schedule requires careful tuning. Moreover, Takida et al. (2022) propose self-annealed stochastic quantization (SQ-VAE) to make the stochastic VQ-VAE converge to deterministic quantization. Unfortunately, the convergence from SQ to VQ relies on the collapse of the likelihood term (i.e., the generative model) (Takida et al., 2022), which is impractical for lossless compression as the likelihood is required for entropy coding.

**Discrete Autoregressive Latent Space** Autoregressive models have been widely applied for image likelihood modeling and generation (van den Oord et al., 2016b;a; Razavi et al., 2019). However, most methods exploit masked convolution (van den Oord et al., 2016b) to implement autoregressive models in the observation space for images, incurring heavy floating-point operations (FLOPs) and increasing the computational cost. Furthermore, the sequential nature of autoregressive models, where each observation depends on previous ones, hinders parallel computations and further diminishes efficiency. Although recent works (Ruan et al., 2021; Ryder et al., 2022; Guo et al., 2022; 2023) develop parallelizable autoregressive models for efficient coding on parallel computation devices like GPUs, their theoretical time complexity remains high, resulting in suboptimal efficiency on general computation devices such as CPUs. One potential solution is to apply autoregressive methods (e.g., masked convolutions) to the lower-dimensional latent space (Minnen et al., 2018; Lee et al., 2018; Cheng et al., 2020), as the latent space is typically smaller than the observation space and subsequently reducing the number of sequential steps. However, the number of FLOPs in masked convolution still remains large.

In this work, our objective is to improve the compression ratio of autoencoder-based codecs while maintaining computational efficiency. To this end, we prefer the discrete autoregressive latent space due to its superior compression ratio, and we aim to further boost its efficiency.

## 3 THE PROPOSED LEARNED LOSSLESS COMPRESSION ARCHITECTURE

In this section, we propose an effective and efficient framework for lossless data compression.

**Overall Architecture** Our proposed lossless compression system combines latent space models with autoregressive models, as depicted in Figure 2. The framework is composed of three main parts: 1) a backbone network that consists of the inference network and generative network, 2) a latent space that incorporates an efficient autoregressive Markov model, and 3) a quantization method for the discrete latent space optimization. In this system, the data is initially processed by an inference

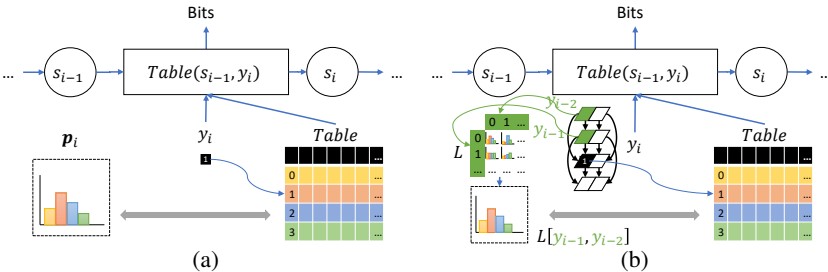

Figure 3: Different Designs of the ANS Coder, where $Table$ represents the state transition lookup table of table ANS (tANS), while $s_{i-1}$ and $s_i$ correspond to the previous and current state of ANS, respectively. Additionally, $p$ denotes the prior probability utilized by ANS to construct $Table$. (a) Non-Autoregressive tANS Coding. (b) Proposed Finite-State Autoregressive (Order-2 Channel) tANS Coding. Here, **L** signifies the lookup table.

network to obtain discrete latent variables **z** using a codebook, which are then quantized to **y** (see Figure 2 (b)) and encoded by an autoregressive latent ANS coder (see Figure 2 (a)). Based on these latent variables, the generative network generates the data likelihood, which is essential for data ANS coding. More details on this framework are provided in Appendix B.

**Backbone Network** The backbone network serves as a crucial component of the latent space model. Generally, deeper networks yield a more compact latent space and better generative results, thus improving the compression ratio. However, deeper networks demand more computational resources, making shallow networks preferable for efficient compression. Moreover, for lossless compression, the generative results should be random rather than deterministic. This is because the distribution of the observation space is necessary for entropy coding of the original data. Thus, autoencoders with collapsed generative networks (Takida et al., 2022) cannot be employed for lossless compression.

**Latent Space** The latent space, serving as a condensed representation of the original data, also needs to be encoded in the final bitstream. As discussed in Section 1, encoding continuous random latent spaces is fraught with difficulties, motivating our preference for efficient frameworks that leverage discrete deterministic latent spaces. In line with previous findings that vector quantized latent spaces yield superior results for generative tasks (Kang et al., 2022), we adopt this format as the latent space representation in our framework. Furthermore, to enhance the compression of the latent space without relying on deep networks as the backbone, we introduce an efficient autoregressive model, namely the FSAR model as a compensatory measure (see Figure 2 (a)), which will be further elaborated upon in Section 4. The FSAR model can be easily implemented with a lookup table and boasts similar time complexity to modern entropy coders such as ANS (Duda, 2009). Moreover, we propose Learnable State Number (LSN) to further reduce the space complexity associated with the lookup table.

**Optimization** As discussed in Section 2, optimizing discrete deterministic latent spaces can be challenging due to the non-differentiable quantization process. Therefore, we analyze existing works and propose an improved quantization scheme, namely STHQ, based on the straight-through estimator (Bengio et al., 2013) (see Figure 2 (b)). This scheme facilitates rapid and robust optimization of the vector quantized latent space, which will be discussed in Section 5. Thanks to the STHQ quantization scheme, the entire framework, encompassing the backbone network, latent space, and vector quantization codebook, can be collectively optimized in an end-to-end manner.

## 4 FINITE-STATE AUTOREGRESSIVE ENTROPY CODING

Autoregressive models have shown promise for improving compression ratios in lossless compression. However, their limited efficiency poses significant restrictions. In this section, we propose a novel and compact approach to autoregressive modeling based on finite-state Markov models. This model acts as the prior for a discretized latent space within deterministic autoencoders, thereby shortening the sequential process and enabling compact coding. Importantly, the implementation of this model using a lookup table showcases low complexity comparable to modern entropy coders. We refer to the combination of FSAR and entropy coder as Finite-State Autoregressive Entropy Coding.

**The Efficiency of Autoregressive Models** Existing autoregressive models are computationally expensive, primarily due to the long sequential process and high time complexity, as detailed in Section 1. On the contrary, modern entropy coders, despite their sequential nature, typically demonstrate satisfactory efficiency due to their low complexity at each step. This suggests that by making autoregressive models sufficiently compact, high computational efficiency is still attainable. Indeed, compression methods prioritizing computational efficiency, such as Ballé et al. (2018) and Zhu et al. (2022), could utilize efficient entropy coders such as tANS (Duda, 2013) on discrete latent

spaces, relying solely on lookup tables with minimal time complexity. Hence, we aim to harness the capabilities of such entropy coders to alleviate the computational burden of autoregressive models.

**Finite-State Markov Models** In a discretized latent space, the latent variables can only take values from a finite set. Consequently, the outcomes of the autoregressive model for a discretized latent space can also be represented by a finite set. To implement such an autoregressive model, we introduce an order-$N$ finite-state Markov model $\mathcal{M}$ that produces prior distributions based on the previous $N$ variables, each with $C$ possible states. Each iteration or step of the autoregressive Markov model can be executed with an $N$-dimensional lookup table $\mathbf{L}$, similar to tANS (Duda, 2013). Denoting samples from the finite set as $\boldsymbol{y}$, this process can be succinctly expressed as:

$$\boldsymbol{p}_i = \mathcal{M}(y_1, ... y_N) = \mathbf{L}[y_1, ... y_N], y_1, ... y_N \in \{1, 2, ..., C\}. \tag{1}$$

As an illustration, Figure 3b shows an order-2 finite-state Markov model combined with tANS coding, leveraging a 2-dimensional lookup table $\mathbf{L}$ to reduce the time complexity. Here, only two memory operations are necessary to read both dimensions of the lookup table and retrieve the corresponding state transition lookup table for tANS.

It is important to mention that the training process of the finite-state Markov model does not explicitly optimize the lookup table. Instead, the transition process in the finite-state Markov model $\mathcal{M}$ can be approximated using neural networks with discrete input, such as a Multi-Layer Perceptron network. Once the training is complete, the lookup table $\mathbf{L}$ can be generated by feeding all possible state combinations into the network.

**Learnable State Number** We notice that the size of the lookup table $\mathcal{O}(C^N)$ can become prohibitively large, especially when $N \geq 2$. A common practice to optimize $C$ is through hyperparameter tuning. However, this tuning process is time-consuming, as it requires training the full model multiple times to determine the optimal hyperparameter. To mitigate this issue, we propose an end-to-end adaptive optimization method for selecting $C$ called Learnable State Number (LSN).

A straightforward approach to adjusting the state number involves applying a masking function to the state probabilities, preventing certain states from being sampled. Additionally, a trainable parameter $\theta$ controlling the sparsity of the mask can be introduced. To facilitate gradient descent, the mask should be a continuous function of $\theta$. Here, we adopt $\alpha$-entmax (Peters et al., 2019), a normalizing function similar to the softmax function, that allows the generation of precisely zero probabilities and enables sparse masking of states. It can be defined as:

$$\mathrm{entmax}(\boldsymbol{\theta}, \alpha) = [(\alpha - 1)\boldsymbol{\theta} - \tau \mathbf{1}]_+^{\frac{1}{\alpha - 1}}. \tag{2}$$

where $\tau$ is a constant that ensures $\sum \mathrm{entmax}(\boldsymbol{\theta}, \alpha) = 1$. The state probabilities, denoted as $\boldsymbol{\pi}$, are then modified by the $\alpha$-entmax function as shown below:

$$\boldsymbol{\pi}' = \frac{\boldsymbol{\pi} \cdot \mathrm{entmax}(\boldsymbol{\theta}, \alpha)}{\sum_C \boldsymbol{\pi} \cdot \mathrm{entmax}(\boldsymbol{\theta}, \alpha)}, \quad \boldsymbol{\pi}, \boldsymbol{\theta} \in \mathbb{R}^C. \tag{3}$$

Note that $\alpha$ can be manually set to control the reduction rate or set as a trainable parameter following (Correia et al., 2019) for more flexible masking during training.

**Discussion for FSAR as Latent Prior** When compressing a natural data vector $\boldsymbol{x}$, it is common for dependencies to exist among the variables in $\boldsymbol{x}$. Markov models are often employed to simplify these dependencies, assuming that each variable depends only on a few preceding variables. However, even with Markov models, decoding each $p(x_i | \boldsymbol{x}_{i-N:i-1})$ still requires the previous $N$ variables, resulting in a sequential iteration process for every element of $\boldsymbol{x}$, which can be time-consuming. Alternatively, latent space models, such as autoencoders, can transform $\boldsymbol{x}$ into a smaller latent vector $\boldsymbol{z}$. By applying Markov models in the latent space, the sequential iteration process can be shortened. Although disentanglement of all latent variables is unattainable under unsupervised optimization targets for lossless compression (Locatello et al., 2018), the inference network in autoencoders can still disentangle certain dependencies among variables, making a compact Markov model sufficient as a prior for the latent space. This approach provides appreciable improvements for compression ratios while preventing overfitting. Consequently, we employ the proposed FSAR model with a low order (typically $N \leq 2$) as the latent prior.

**Relationship to Masked Convolution (van den Oord et al., 2016b)** Masked convolution can also be exploited as an autoregressive prior in the discrete latent space. In fact, the masked convolution-based autoregressive model is essentially a subset of the finite-state Markov model. However, constructing the lookup table becomes nearly impossible due to the high order in most cases. For instance, a 3x3 2D

masked convolution-based autoregressive model corresponds to an Order-4 finite-state Markov model in the discrete latent space, denoted as $p(y_{i,j,k}|y_{i,j,k-1}, y_{i,j-1,k-1}, y_{i,j-1,k}, y_{i,j-1,k+1})$, where $i, j, k$ represent the indices of the channel, height, and width dimensions, respectively. Generating the lookup table for this model requires a space complexity of $\mathcal{O}(C^4)$, which is generally unacceptable. Furthermore, mask convolution usually requires FLOPs, which can introduce potential inconsistencies in computation between the compression and decompression stages Ballé et al. (2019). On the contrary, the proposed method is immune to such inconsistencies as it relies solely on memory operations for latent coding.

**Relationship to ANS (Duda, 2009)** Recent implementations of entropy coders have embraced ANS (Duda, 2009), as shown in Figure 3a. In the $i$-th step, ANS utilizes the previous state $s_{i-1}$, the current symbol $y_i$, and the corresponding prior distribution $p$ to determine the state transition process. When the inputs of the state transition process, namely $s_{i-1}$ and $y_i$, are drawn from a finite set, techniques such as tANS (Duda, 2013) can be employed to precalculate all possible inputs and construct a lookup table for efficient state transition computation. This concept inspires us to devise a similar scheme for coding in the discrete latent space. In fact, the proposed FSAR can seamlessly integrate with tANS, where a state transition table is derived from the FSAR lookup table **L** at each step (see Figure 3b). For a detailed implementation of FSAR-tANS and a discussion on the time complexity of related methods, we direct the readers to Appendix C.

# 5 STRAIGHT THROUGH HARDMAX QUANTIZATION

As mentioned in Section 1, optimizing discrete latent space models is complicated despite their preference for efficient latent coding. In this section, we examine the limitations of existing methods for optimizing vector quantized (VQ) discrete latent space models and propose a robust quantization method called Straight Through Hardmax Quantization (STHQ).

Recall that the original VQ-VAE (van den Oord et al., 2017) is optimized with a K-means-like loss:
$$\mathcal{L}_{VQ} = ||\operatorname{sg}(I(\boldsymbol{x})) - \boldsymbol{B}\boldsymbol{y}^T||_2 + \beta||I(\boldsymbol{x}) - \operatorname{sg}(\boldsymbol{B}\boldsymbol{y}^T)||_2 - \log p(\boldsymbol{x}|\boldsymbol{B}\boldsymbol{y}^T), \tag{4}$$
where $I(\boldsymbol{x})$ is the output of the inference network, $\boldsymbol{B}$ represents the codebook, $\boldsymbol{y}$ denotes the one-hot vector that selects the closest codeword to $I(\boldsymbol{x})$ in $\boldsymbol{B}$, and sg represents the stop gradient operator. However, updating all codewords in each step is infeasible due to the one-hot selection of a single codeword by $\boldsymbol{y}$, resulting in zero gradients for other codewords in the codebook (Sønderby, 2017; Roy et al., 2018).

To overcome this limitation, the relaxed VQ (RVQ) approach (Sønderby, 2017) introduces a categorical distribution $q(\boldsymbol{z}|\boldsymbol{x})$ in the latent space. The logits of this distribution are determined by the Euclidean distance between the latent variable and the codewords, that is,
$$q(\boldsymbol{z}|\boldsymbol{x}) = \operatorname{Categorical}(\operatorname{softmax}(-\boldsymbol{D}(I(\boldsymbol{x})))), \tag{5}$$
where $\boldsymbol{D}_i(I(\boldsymbol{x})) = ||I(\boldsymbol{x}) - \boldsymbol{B}_i||_2$ is the euclidean distance between $I(\boldsymbol{x})$ and $i$-th codeword vector $\boldsymbol{B}_i$. The entire model is then optimized using the negative evidence lower bound (ELBO) formulation:
$$\mathcal{L}_{RVQ} = -H(q(\boldsymbol{z}|\boldsymbol{x})) - \mathbb{E}_q(\log p(\boldsymbol{z})) - \log p(\boldsymbol{x}|\boldsymbol{z}), \tag{6}$$
where $H(q(\boldsymbol{z}|\boldsymbol{x}))$ is the entropy of $q(\boldsymbol{z}|\boldsymbol{x})$. The inclusion of the entropy term facilitates global optimization of all distances $\boldsymbol{D}$, encouraging close latent variables and codewords to be grouped together. This formulation has been adopted by recent works such as SQ-VAE (Takida et al., 2022).

However, there are two potential issues immediately arising when optimizing the entropy term in RVQ. Firstly, the partial derivative of the entropy term with respect to the distance $\boldsymbol{D}_i$ can be negative for all $i$, which could be derived from Proposition 5.1. As a consequence, the distance may continually increase during the optimization process and cause the codebook to diverge.

**Proposition 5.1.** *It is possible that $\forall i, \partial H(q(\boldsymbol{z}|\boldsymbol{x}))/\partial \boldsymbol{D}_i \leq 0$.*

*Proof.* See Appendix D.1 for proof. □

Secondly, the inclusion of the entropy term $H(q(\boldsymbol{z}|\boldsymbol{x}))$ creates an entropy gap between VQ and RVQ, resulting in an inferior performance of RVQ in generative tasks, as proven in Proposition 5.2:

**Proposition 5.2.** *Assume $p(z)$ follows a uniform distribution and give the same sample input $\boldsymbol{y}$ and parameters $\boldsymbol{\theta}$ for the generative model $\log p_{\boldsymbol{\theta}}(\boldsymbol{x}|\boldsymbol{B}\boldsymbol{y}^T)$, we have $\log p_{\arg\min_{\boldsymbol{\theta}} \mathcal{L}_{RVQ}}(\boldsymbol{x}|\boldsymbol{B}\boldsymbol{y}^T) < \log p_{\arg\min_{\boldsymbol{\theta}} \mathcal{L}_{VQ}}(\boldsymbol{x}|\boldsymbol{B}\boldsymbol{y}^T)$.*

*Proof.* See Appendix D.2 for proof and see Appendix D.4 for the corresponding experiments. □

In order to avoid these problems and enable global updates similar to RVQ, we propose STHQ.

**Straight Through Hardmax Quantization** Inspired by the straight through Gumbel-softmax method (Jang et al., 2016) which employs categorical one-hot samples in the forward process but Gumbel-softmax based soft samples in the backward process (backpropagation), we propose a similar gradient estimator called straight through hardmax ($\mathrm{st\text{-}hardmax}$). This approach incorporates a VQ-based *hardmax* in the forward process while using sampling in the backward process:

$$\boldsymbol{y} = \mathrm{st\text{-}hardmax}(-\boldsymbol{D}(I(\boldsymbol{x}))) \leftrightarrow \boldsymbol{y} = \mathrm{hardmax}(-\boldsymbol{D}(I(\boldsymbol{x}))), \ \frac{\partial \boldsymbol{y}}{\partial \boldsymbol{D}} \overset{ST}{=} \frac{\partial \mathrm{GS}_\tau(-\boldsymbol{D}(I(\boldsymbol{x})))}{\partial \boldsymbol{D}}, \ (7)$$

where $\mathrm{GS}_\tau$ represents the Gumbel-softmax reparameterization (Jang et al., 2016) with temperature $\tau$. In this manner, the forward process remains the same as VQ, thereby mitigating the entropy gap issue. Meanwhile, the backward process allows for the global update of $\boldsymbol{z}$ and all $\boldsymbol{B}_i$ based on their respective distances $\boldsymbol{D}_i$. Consequently, we can use a simple regularization loss term $||\boldsymbol{D}_{min}||_2$, similar to VQ-VAE, to update all distances $\boldsymbol{D}i$:

$$||\boldsymbol{D}_{min}||_2 = || \arg\min_i \boldsymbol{D}_i ||_2 = ||I(\boldsymbol{x}) - \boldsymbol{B}\boldsymbol{y}^T||_2. \tag{8}$$

By combining the $\mathrm{st\text{-}hardmax}$ operator and this L2 regularization term, the overall loss function of STHQ can be expressed as:

$$\mathcal{L}_{STHQ} = ||\boldsymbol{D}_{min}||_2 - \log p(\boldsymbol{x}|\boldsymbol{B}\boldsymbol{y}^T) - \mathbb{E}_q \log p(\boldsymbol{z}). \tag{9}$$

Note that in VQ, $p(\boldsymbol{z})$ is typically assumed to follow a uniform distribution and is omitted from the loss function. However, in STHQ, we adopt the proposed FSAR model as $p(\boldsymbol{z})$ for better compression. To learn the parameters of the FSAR model, $p(\boldsymbol{z})$ needs to be included in Eq. (9). Specifically, following previous VQ methods, we only utilize $\mathbb{E}_q \log p(\boldsymbol{z})$ for optimizing the FSAR model, while excluding this term when optimizing the backbone network and the codebook.

**Relationship to Other VQ-based Methods** By assuming $p(\boldsymbol{z})$ to be uniform in STHQ (9) and ignoring the stop-gradient operations in VQ-VAE (4), we can find that the loss functions in these two methods are identical. It is worth noting that both VQ-VAE and STHQ apply the straight-through estimator on the likelihood term $\log p(\boldsymbol{x}|\boldsymbol{B}\boldsymbol{y}^T)$ to enable gradient calculation on the inference model. However, in STHQ, we also apply the straight-through estimator to the L2 regularizer term $||\boldsymbol{D}_{min}||_2$, which benefits from the global update property and leads to faster convergence. On the other hand, when compared to RVQ-based methods such as (Sønderby, 2017; Takida et al., 2022), STHQ omits the entropy term. This term can potentially hinder the optimization process, as indicated in the above two propositions. More details are presented in Appendix D.3.

## 6 EXPERIMENTS

### 6.1 DATASETS AND METRICS

In our experiments, we focused on compressing and decompressing image datasets, specifically CIFAR10 (CF10) (Krizhevsky, 2009) and ImageNet32/64 (IN32, IN64) (Deng et al., 2009). We evaluated performance using four criteria: Bits Per Dimension (BPD) for compression ratio, compression speed (CSpd) and decompression speed (DSpd) measured in megabytes per second (MB/s) to assess time complexity, and occupied memory (Mem) in megabytes (MB) to assess space complexity. The measured time or speed was obtained running on CPUs, as our method targets general-purpose computation devices. The detailed experimental setup can be found in Appendix F.1.

### 6.2 COMPARISON WITH OTHER LATENT CODING METHODS

To demonstrate the effectiveness and efficiency of the proposed FSAR model as a prior for latent modeling, we conducted a comparative analysis with various other latent space coding methods. The methods compared include commonly used continuous latent models with bits-back coding (Townsend et al., 2019a), discrete latent models such as Vector Quantization (VQ) (van den Oord et al., 2017), Universal Quantization (UQ) (Ballé et al., 2018), and McQuic (Zhu et al., 2022), as well as autoregressive-based latent models such as Masked Convolution (MaskConv) (van den Oord et al., 2016b) and Checkerboard context model (Checkerboard) (He et al., 2021). Our proposed FSAR model encompassed Markov models of different orders, ranging from Order-1 to Order-3. Additionally, we consider the Order-2 model with LSN to evaluate its usefulness. We utilized the proposed STHQ training for all autoregressive methods (MaskConv2D, Checkerboard, FSAR(O1), FSAR(O2), FSAR(O2)+LSN, FSAR(O3)), while also conducting an additional experiment in the non-autoregressive latent space (i.e., only tANS (Duda, 2013) entropy coding) as a reference. Furthermore, to provide detailed efficiency comparisons, we presented the average decompression time of the

Table 1: Experiment results of different latent coding methods. The detailed running time for each module in different methods is obtained by processing a single image from CIFAR10 dataset. Order-1, Order-2 and Order-3 denoted as O1, O2, and O3, respectively. For BPD results, bold numbers indicate the best practical BPD for each dataset, "OOM" indicates the out-of-memory error during coding, "ERR" indicates inconsistency between the decompressed data and the original data, and "NAN" indicates a failed training process.

| Methods | | BPD (Theoritical/Practical) | | | CF10 Decomp Time (ms) | | | Mem (MB) |
|---|---|---|---|---|---|---|---|---|
| | | CF10 | IN32 | IN64 | Net | Coder | Total | CF10 |
| Continuous | bits-back (Townsend et al., 2019a) | 4.99/8.99 | 5.45/9.11 | 5.05/11.15 | 32.44 | 9.85 | 42.29 | 0.000 |
| Discrete | UQ (Ballé et al., 2018) | 5.16/5.19 | 5.65/5.68 | 5.36/5.38 | 1.60 | 1.54 | 3.15 | 0.015 |
| | VQ (van den Oord et al., 2017) | 5.50/5.52 | 6.16/6.17 | 5.61/5.62 | 1.57 | 0.88 | 2.45 | 0.008 |
| | McQuic (Zhu et al., 2022) | 5.81/5.50 | 6.19/9.57 | NAN/NAN | 1.57 | 8.75 | 10.32 | 0.082 |
| Autoregressive | MaskConv2D (van den Oord et al., 2016b) | 5.02/ERR | 5.42/ERR | 5.08/ERR | 1.75 | 632.24 | 633.99 | 1.127 |
| | Checkerboard (He et al., 2021) | 5.11/5.11 | 5.49/5.50 | 5.23/5.23 | 1.63 | 4.71 | 6.34 | 1.127 |
| Proposed | tANS (Duda, 2013) | 5.25/5.25 | 5.72/5.72 | 5.46/5.48 | 1.66 | 0.86 | 2.52 | 0.008 |
| | FSAR(O1) | 5.14/5.17 | 5.54/5.55 | 5.16/5.18 | 1.58 | 0.97 | 2.55 | 2.010 |
| | FSAR(O2) | 5.05/**5.06** | 5.45/**5.46** | 5.05/**5.07** | 1.62 | 1.04 | 2.65 | 516.512 |
| | FSAR(O2)+LSN | 5.08/5.09 | 5.43/**5.46** | 5.10/5.12 | 1.59 | 1.06 | 2.65 | 276.586 |
| | FSAR(O3) | 5.00/OOM | 5.41/OOM | 5.08/OOM | OOM | OOM | OOM | OOM |

backbone networks (Net) and the entropy coders (Coder) for processing a single image in CIFAR10. The results are shown in Table 1. Compression time are further provided in Appendix E.1 for reference. It is apparent that the proposed method achieves comparable or superior performance to the existing methods in terms of both compression ratio (i.e., theoretical/practical BPD) and speed (i.e., run time).

Concretely, in terms of the practical BPD, the proposed FSAR-based methods achieve the best performance on all datasets. Moreover, the theoretical BPD closely aligns with the practical one for all methods, except for bits-back and MaskConv. This discrepancy arises from the initial bits issue encountered in bits-back (Flamich et al., 2020) and the floating-point inconsistency issue observed in MaskConv (Ballé et al., 2019). Comparing FSAR models of different orders, we find that the order-2 Markov model outperforms the order-1 model in terms of compression performance, while the order-3 model does not provide a significant improvement over the order-2 model. This observation implies that a compact Markov model can adequately describe the latent space, as the backbone model is capable of disentangling dependencies within the data to some extent, resulting in a simplified dependency structure in the latent space. Furthermore, considering the memory-intensive nature of order-3 models, the order-2 Markov model is the optimal choice for FSAR.

In terms of speed, the bits-back and MaskConv methods are notably slow due to their high time complexity in the backbone network and entropy coder, respectively. The Checkerboard context model employs parallelized computation during latent decoding, resulting in significantly shorter decoding time. However, Checkerboard remains slower than the non-autoregressive tANS, whereas the FSAR model performs as fast as tANS. In addition, the increased time cost of accessing lookup tables in FSAR, which scales with memory size, has minimal impact on the overall decompression time, as the time is dominated by the FLOPs in the backbone networks.

Note that memory occupation is a concern for FSAR models. Although the proposed LSN method successfully reduces the memory requirement of FSAR(O2) by approximately 50% without causing noticeable changes in compression performance, the memory occupation remains relatively high.

## 6.3 Comparison with Other Compression Methods

In this section, we evaluate the proposed codec by comparing it with state-of-the-art image compression methods in a general setting. No restrictions are imposed on the network backbone or training schedule in this experiment. The benchmark methods include traditional image lossless compression codecs such as PNG (Boutell, 1997), WebP (webmproject, 2023) and FLIF (Sneyers & Wuille, 2016), as well as autoencoder based methods including L3C (Mentzer et al., 2018), bitswap (Kingma et al., 2019), and PILC (Kang et al., 2022). Since the proposed method solely employs latent space models and does not restrict the method of observation space data coding, it can be seamlessly combined with PILC, which adopts autoregressive models and entropy coding in the observation space, by replacing the VQ-VAE used in PILC with the proposed method. We calculate the BPD based on the length of the compression bitstream to evaluate practical compression performance. For a qualitative understanding of the results, we present the BPD versus Speed comparison for different methods on CIFAR10 in Figure 1. For complete numerical results, please refer to Appendix E.2.

Generally, the proposed learned codec has similar BPD and speed with traditional WebP and FLIF, thereby bridging the gap between learned and traditional methods. Compared to the autoencoder-

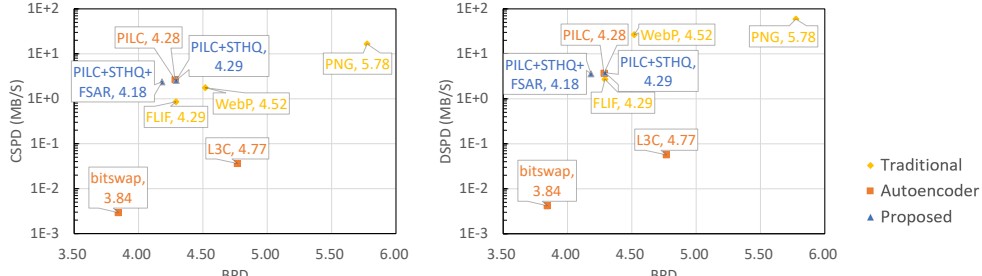

Figure 4: The BPD versus Speed (CSpd and Dspd) comparison for different methods on CIFAR10.

Table 2: BPD results (with standard deviation) for the ablation studies.

| | BPD | Saved | | BPD | Saved | | BPD | Saved |
|---|---|---|---|---|---|---|---|---|
| VQ (van den Oord et al., 2017) | 5.37(0.02) | 0.00% | STHQ | 5.23(0.02) | -2.66% | -STH | 5.54(0.08) | 3.16% |
| RVQ (Sønderby, 2017) | 5.68(0.10) | 5.73% | +FSAR(O1) | 5.15(0.03) | -4.16% | -L2 | 6.58(0.03) | 22.36% |
| SQ (Takida et al., 2022) | 5.50(0.11) | 2.42% | +FSAR(O2) | 5.06(0.02) | -5.80% | +Entropy | 5.34(0.06) | -0.67% |

based method PILC, the improved version of the proposed method, called "PILC+STHQ+FSAR," reduces BPD by 2.2-2.5% but also slightly decreases CSpd by 7% and DSpd by 5%. The reduced BPD can be attributed to the incorporation of the FSAR model. Additionally, the compression performance of our method is similar to that of the deterministic autoencoder-based method L3C (Mentzer et al., 2018), but is inferior to the variational autoencoder-based method bitswap (Kingma et al., 2019). However, in terms of speed, our method outperforms both L3C and bitswap by a large margin, owing to the shallow backbone and the efficient entropy coder. It is worth noting that bitswap requires initial bits, rendering it unsuitable for single or small-batch image compression.

## 6.4 Ablation Studies

Here, we conduct ablation studies to examine the role of each component in the proposed method. The theoretical BPD is reported, as previous experiments have shown a negligible disparity between theoretical and practical compression performance. Our baseline is the widely-used VQ-VAE with a discrete latent space (van den Oord et al., 2017). We compare it with RVQ-VAE (Sønderby, 2017), SQ-VAE (Takida et al., 2022), and the proposed STHQ-based methods. For a fair comparison, we disable the self-annealing mechanism in SQ-VAE for the sake of lossless compression. Different variants of STHQ are considered, including the one with the FSAR prior (+FSAR), without the st-hardmax operator (-STH), without the L2 regularization term $||\boldsymbol{D}_{min}||_2$ (-L2), and with an entropy term similar to RVQ-VAE (+Entropy). The experiments are performed on CIFAR10 images using the same backbone network and sizes of latent variables and codebooks. The results are presented in Table 2. For additional results from the ablation study, including different settings of the backbone network, latent space, and codebook size, we refer the readers to Appendix E.3-E.5.

It is obvious that the proposed method surpasses all variants. STHQ reduces the BPD by about 2% compared to VQ-VAE, and the inclusion of FSAR yields an additional 2-4% improvement. RVQ-VAE and SQ-VAE perform worse than VQ-VAE, probably due to the entropy gap discussed in Section 5. The higher variance observed in RVQ-VAE and SQ-VAE suggests instability, which can be explained by the codebook divergence (cf. Proposition 5.1). Regarding the variants of STHQ, we observe that the L2 regularization term achieves a 20% BPD decrease by supporting the simultaneous update of all codewords. Moreover, without the st-hardmax operator, the distance updates in STHQ resemble those of SQ-VAE, and so both STHQ-st-hardmax and SQ-VAE perform similarly but worse than STHQ by 6%. Recall that SQ-VAE is unsuitable for compression purposes due to its reliance on likelihood degeneration to obtain a deterministic decoder (see Section 2). In addition, introducing the entropy term can cause instability in the optimization process (cf. Proposition 5.1), thus degrading STHQ's performance by about 2%. In summary, the st-hardmax operator, L2 regularization term, and exclusion of the entropy term are all vital for the robust optimization of the discrete latent space.

## 7 Conclusions

This paper presents a novel framework for efficient learned lossless compression, which incorporates FSAR for efficient autoregressive latent coding and improved compression, together with STHQ for robust optimization of the discrete latent space and further enhancement of compression performance. Experimental results demonstrate a substantial improvement over the baseline, with a 6% increase in compression ratio while maintaining comparable compression and decompression speeds. These findings highlight the potential of our approach for advancing the field of lossless compression.

ACKNOWLEDGMENTS

The work was supported by Ant Group and the National Natural Science Foundation of China (No. 62325109, U21B2013).

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

## A    FURTHER REVIEW ON RELATED WORKS

**Continuous Random Latent Space** Variational autoencoders (VAE) with continuous random latent space (Kingma & Welling, 2013) are commonly employed in learned lossless compression due to their ease of optimization using the Evidence Lower Bound (ELBO). However, the practical encoding of their latent space poses a challenge, as the random samples are continuous and cannot be directly processed by a simple entropy coder. To tackle this issue, previous approaches have proposed two main solutions: bits-back based latent coding (Townsend et al., 2019a; Kingma et al., 2019; Theis & Ho, 2021; Ruan et al., 2021) and relative entropy coding (Flamich et al., 2020; 2022). Bits-back ANS (Townsend et al., 2019a) suggests taking random samples directly from the compressed stream. Subsequent works have extended or optimized bits-back for different types of VAE variants, including hierarchical VAEs (Townsend et al., 2019b; Kingma et al., 2019), importance weighted autoencoders (Theis & Ho, 2021), and Monte-Carlo VAEs (Ruan et al., 2021). Another approach, relative entropy coding (Flamich et al., 2020; 2022), incorporates a shared random number generator between the compressor and decompressor, compressing the sampled index to represent latent samples. However, both methods still require additional bits for the random source, and the implementation of the entropy coder remains complex.

## B    BACKGROUND KNOWLEDGE FOR SECTION 3

**Entropy Coding with ANS** Entropy coding plays a central role in most compression frameworks, as it compresses messages sampled from a probability distribution into a bitstream with a length close to its entropy (Shannon, 1948), expressed as $-\mathbb{E}_P \log_2 P(\boldsymbol{x})$. For a single sample from $P(\boldsymbol{x})$, the optimal code length is $-\log_2 P(\boldsymbol{x})$. A recently proposed near-optimal entropy coder, ANS (Duda, 2009), treats the bitstream as a growing state. It pushes and pops values to and from this state based on their probabilities. One of the ANS implementations, tabled ANS, is illustrated in Figure 3a. ANS achieves the optimal code length $-\log_2 P(\boldsymbol{x})$ for any given samples from $P(\boldsymbol{x})$.

**Compression with Variational Autoencoders** The proposed framework draws inspiration from previous compression frameworks based on variational autoencoders. A variational autoencoder models the probability of the input data $p(\boldsymbol{x})$ by transforming $\boldsymbol{x}$ to latent variables $\boldsymbol{z}$ and generating the likelihood $p(\boldsymbol{x}|\boldsymbol{z})$ based on $\boldsymbol{z}$. Consequently, $\boldsymbol{x}$ can be compressed or decompressed using $p(\boldsymbol{x}|\boldsymbol{z})$ by applying an entropy coder. The latent variables $\boldsymbol{z}$ follow a random distribution conditioned on $\boldsymbol{x}$, denoted as $q(\boldsymbol{z}|\boldsymbol{x})$. As $\boldsymbol{z}$ is required by $p(\boldsymbol{x}|\boldsymbol{z})$, $\boldsymbol{z}$ also needs to be entropy coded with $p(\boldsymbol{z})$. The entire process of modeling $p(\boldsymbol{x})$ using a variational autoencoder can be represented as:

$$p(\boldsymbol{x}) = \frac{p(\boldsymbol{z})p(\boldsymbol{x}|\boldsymbol{z})}{q(\boldsymbol{z}|\boldsymbol{x})}. \tag{10}$$

The length of the compressed stream produced by entropy coders can be estimated using $-\mathbb{E}_q \log p(\boldsymbol{x})$, which can be rewritten as:

$$-\mathbb{E}_q \log p(\boldsymbol{x}) = \mathbb{E}_q \log q(\boldsymbol{z}|\boldsymbol{x}) - \log p(\boldsymbol{z}) - \log p(\boldsymbol{x}|\boldsymbol{z}) = KL(q(\boldsymbol{z}|\boldsymbol{x})||p(\boldsymbol{z})) - \log p(\boldsymbol{x}|\boldsymbol{z}). \tag{11}$$

Here, $-\log p(\boldsymbol{x}|\boldsymbol{z})$ estimates the compressed length of $\boldsymbol{x}$, and $KL(q(\boldsymbol{z}|\boldsymbol{x})||p(\boldsymbol{z}))$ estimates the compressed length of $\boldsymbol{z}$. It's important to note that common entropy coders can only process samples rather than distributions. Hence, the true compressed length should satisfy the inequality given by:

$$-\log p(\boldsymbol{z}) - \log p(\boldsymbol{x}|\boldsymbol{z}) \geq \mathbb{E}_q \log q(\boldsymbol{z}|\boldsymbol{x}) - \log p(\boldsymbol{z}) - \log p(\boldsymbol{x}|\boldsymbol{z}) = -\mathbb{E}_q \log p(\boldsymbol{x}). \tag{12}$$

As $q(\boldsymbol{z}|\boldsymbol{x})$ is a discrete distribution, equality is achieved only if $q(\boldsymbol{z}|\boldsymbol{x})$ is a one-hot distribution. Therefore, we can straightforwardly set $q(\boldsymbol{z}|\boldsymbol{x})$ as a categorical distribution with one-hot probabilities, which can be easily parameterized using a quantization function.

As shown in Figure 2, the inference network provides parameters for $q(\boldsymbol{z}|\boldsymbol{x})$, while the generative network implements $p(\boldsymbol{x}|\boldsymbol{z})$. The one-hot representation of $q(\boldsymbol{z}|\boldsymbol{x})$ is obtained by quantizing the output of the inference network using a learnable codebook, employing the proposed STHQ method. To be specific, we calculate the euclidean distances between the inference network output and each codeword in the learnable codebook. The negative euclidean distances are then utilized to determine the sampling probability of each codeword, following a similar approach as in (Sønderby, 2017). Additionally, we incorporate the sparse mask from the LSN module in FSAR to further prune the codebook. The proposed STHQ produces one-hot latent samples by selecting the codeword with the highest probability through quantization. These one-hot samples can be converted into latent indices for compression into the latent stream and are also used in the dequantization process, where

a specific codeword is chosen from the codebook as the input for the generative network. $p(\boldsymbol{z})$ is effectively encoded using an efficient autoregressive entropy coder based on the proposed FSAR model. The optimization process can use Eq. (12) as the loss function to minimize the length of the compressed bitstream and achieve superior compression ratios.

## C    DISCUSSIONS FOR SECTION 4

### C.1    TIME COMPLEXITY COMPARISON

**Comparison of Complexity with ANS**    The utilization of lookup tables has been explored in table ANS (tANS) implementation (Duda, 2013). This approach precalculates all possible state transitions and utilizes lookup tables during the coding process to expedite the process. Given that the autoregressive Markov model can be iterated alongside the entropy coder, it is straightforward to design a similar lookup table-based autoregressive Markov model for acceleration. To compare the complexity, we provide a breakdown of the number of operations for different ANS implementations, including range ANS (rANS) (Duda, 2013) and table ANS (tANS) (Duda, 2013), as shown in Table 3. It is evident that the proposed FSAR model requires only a few additional memory operations compared to ANS, thanks to its lookup table-based iteration process.

**Comparison of Complexity with Masked Convolution based Autoregressive Models**    Masked convolution (van den Oord et al., 2016b), which relies on FLOPs, is commonly employed for implementing autoregressive models. In Table 4, we provide a breakdown of the number of operations per iteration required for masked convolution-based autoregressive models. In contrast, the proposed FSAR model completely avoids expensive FLOPs and exhibits complexity comparable to that of entropy coders, as shown in Table 3.

Table 3: Integer, floating-point, and memory operations per symbol in modern entropy coders. Operations are distinguished by color-coded boxes in the accompanying pseudo-code.

| | PseudoCode | Int | Float | Mem |
|---|---|---|---|---|
| rANS Encode | start, freq = CDF $[y_i]$ ;
s = $(s_{i-1}$ / freq) » nbit + $s_{i-1}$ % freq + start;
$s_i =$ flushbits (s); | 6 | 0 | 1 |
| rANS Decode | cfreq = $s_{i-1}$ & bitmask;
$y_i$, start, freq = ICDF [cfreq] ;
s = freq * $(s_{i-1}$ » nbit) + cfreq - start;
$s_i =$ readbits (s) | 6 | 0 | 1 |
| tANS Encode | nbit, delta = TransTable $[y_i]$ ;
flushbits $(s_{i-1})$;
$s_i =$ StateTable $[(s_{i-1}$ » nbit) + delta]$ | 3 | 0 | 2 |
| tANS Decode | nbit, $y_i$, s = DecTable $[s_{i-1}]$ ;
$s_i =$ readbits (s) | 1 | 0 | 1 |
| FSAR(O2)+
tANS Encode | TransTable, StateTable = Table $[y_{j-1}]$ $[y_{k-1}]$ ;
tANSEncode (TransTable, StateTable); | 3 | 0 | 4 |
| FSAR(O2)+
tANS Decode | DecTable = Table $[y_{j-1}]$ $[y_{k-1}]$ ;
tANSDecode (DecTable); | 1 | 0 | 3 |

Table 4: Integer, floating-Point, and memory operations per symbol in various autoregressive models. Operations are distinguished by color-coded boxes in the accompanying formula or pseudo-code. Convolutional layers are assumed to have a single channel. $\mathbb{M}a,b$ and $\mathbb{C}a,b$ represent kernel sets, defined as $\mathbb{M}a,b = (j,k)|\forall j,k \in \mathbb{Z}, -a < j < 0, |k| < b \cup (0,k)|\forall k < 0$ and $\mathbb{C}a,b = (j,k)|\forall j,k \in \mathbb{Z}, (j+k) \mod 2 \neq 0$, respectively.

| | Formula/PseudoCode | Int | Float | Mem |
|---|---|---|---|---|
| MaskConv2D3x3 | $\sum_{(m,n)\in\mathbb{M}_{2,2}}^{m,n} y_{j+m,k+n}W_{m,n}$ | 0 | $2|\mathbb{M}_{2,2}| = 8$ | 0 |
| MaskConv2D5x5 | $\sum_{(m,n)\in\mathbb{M}_{4,4}}^{m,n} y_{j+m,k+n}W_{m,n}$ | 0 | $2|\mathbb{M}_{4,4}| = 24$ | 0 |
| Checkerboard2D3x3 | $\sum_{(m,n)\in\mathbb{C}_{2,2}}^{m,n} y_{j+m,k+n}W_{m,n}$ | 0 | $2|\mathbb{C}_{2,2}| = 8$ | 0 |
| FSAR(O1) | Table $[y_{k-1}]$ | 0 | 0 | 1 |
| FSAR(O2) | Table $[y_{j-1}]$ $[y_{k-1}]$ | 0 | 0 | 2 |

## C.2 IMPLEMENTATION OF FSAR-TANS

In theory, FSAR has the potential to be combined with any entropy coders. However, for practical implementation efficiency, we have opted for tANS (Duda, 2013), which leverages lookup tables for accelerated processing. The algorithm for finite-state autoregressive entropy coding based on tANS, including the initialization, encoding, and decoding processes, is illustrated in Algorithm 1. The modified or added steps, which differ from the original tANS algorithm, are highlighted in red. It is obvious that the only additional step during encoding and decoding is the $n$-dimensional table lookup.

## D DISCUSSIONS FOR SECTION 5

### D.1 PROOF OF PROPOSITION 5.1

Proposition 5.1 It is possible that $\forall i, \partial H(q(\boldsymbol{z}|\boldsymbol{x}))/\partial \boldsymbol{D}_i \leq 0$

*Proof.*

$$\frac{\partial H(q(\boldsymbol{z}|\boldsymbol{x}))}{\partial \boldsymbol{D}_i} = -\frac{\partial \sum_i q_i \log q_i}{\partial q_i} \frac{\partial q_i}{\partial \boldsymbol{D}_i}. \tag{13}$$

Note that

$$q_i = \text{softmax}_i(-\boldsymbol{D}) = \frac{e^{-\boldsymbol{D}_i}}{\sum_i e^{-\boldsymbol{D}_i}}. \tag{14}$$

Let $\boldsymbol{S}_i = e^{-\boldsymbol{D}_i} \in (0,1]$, we have

$$\frac{\partial \sum_i (q_i \log q_i)}{\partial q_i} = 1 + \log q_i = 1 + \log \boldsymbol{S}_i - \log \sum_i \boldsymbol{S}_i, \tag{15}$$

and

$$\frac{\partial q_i}{\partial \boldsymbol{D}_i} = \frac{\partial \boldsymbol{S}_i}{\partial \boldsymbol{D}_i} \frac{\partial q_i}{\partial \boldsymbol{S}_i} = -\boldsymbol{S}_i \frac{\sum_i \boldsymbol{S}_i - \boldsymbol{S}_i}{(\sum_i \boldsymbol{S}_i)^2}. \tag{16}$$

Therefore,

$$\frac{\partial H(q(\boldsymbol{z}|\boldsymbol{x}))}{\partial \boldsymbol{D}_i} = \boldsymbol{S}_i(1 + \log \boldsymbol{S}_i - \log \sum_i \boldsymbol{S}_i)\frac{\sum_i \boldsymbol{S}_i - \boldsymbol{S}_i}{(\sum_i \boldsymbol{S}_i)^2}. \tag{17}$$

Since

$$\boldsymbol{S}_i\frac{\sum_i \boldsymbol{S}_i - \boldsymbol{S}_i}{(\sum_i \boldsymbol{S}_i)^2} > 0, \tag{18}$$

---

**Algorithm 1** Finite-State Autoregressive tANS Coding Algorithm. The steps that are different from tANS are highlighted in red.

---

**procedure** INITIALIZATION(FSMarkov $\mathcal{M}$, NumStates $C$, Order $n$)
    Initialize $n$-dimensional Lookup Table $\mathbf{L_E}, \mathbf{L_D}$
    **for** $y_1 \in \{0, 1, ..., C\}, ..., y_n \in \{0, 1, ..., C\}$ **do**
        $\boldsymbol{\pi} \leftarrow \mathrm{softmax}(\mathcal{M}(y_1, ..., y_n))$
        $\mathbf{L_E}[y_1, ..., y_n] \leftarrow BuildTansEncodeTable(vpi)$
        $\mathbf{L_D}[y_1, ..., y_n] \leftarrow BuildTansDecodeTable(\boldsymbol{\pi})$
    **end for**
    **return** $\mathbf{L_E}, \mathbf{L_D}$
**end procedure**
**procedure** ENCODE(Symbols $\boldsymbol{y}$, EncoderLookupTable $\mathbf{L_E}$, Offsets $\boldsymbol{o} = \{o_1, ..., o_n\}$)
    Initialize ANSState $s$, BitStream $B$
    **for** $y_i \in reserve(\boldsymbol{y})$ **do**                        ▷ ANS requires a LIFO queue
        $\mathbf{L} \leftarrow \mathbf{L_E}[y_{i-o_1}, ..., y_{i-o_n}]$
        $T \leftarrow GetEncodeStateTransition(\mathbf{L}, y_i)$
        $s, nbits \leftarrow T(s)$
        $OutputBits(B, LowerBits(s, nbits))$
        $s \leftarrow s >> nbits$
    **end for**
    $Flush(B, s)$
    **return** $B$
**end procedure**
**procedure** DECODE(BitStream $B$, DecoderLookupTable $\mathbf{L}_D$, Offsets $\boldsymbol{o} = \{o_1, ..., o_n\}$)
    Initialize Symbols $\boldsymbol{y}$
    $i \leftarrow 0$
    $s \leftarrow InputBits(B, MaxBits(s))$
    **while** not $IsFinished(B)$ **do**
        $\mathbf{L} \leftarrow \mathbf{L}_D[y_{i-o_1}, ..., y_{i-o_n}]$
        $T \leftarrow GetDecodeStateTransition(\mathbf{L}, s)$
        $y_i, nbits \leftarrow T(s)$
        $s \leftarrow (s << nbits) + InputBits(B, nbits)$
        $i \leftarrow i + 1$
    **end while**
    **return** $\boldsymbol{y}$
**end procedure**

---

to guarantee that $\partial H(q(\boldsymbol{z}|\boldsymbol{x}))/\partial \boldsymbol{D}_i > 0$, the following inequality must hold:

$$1 + \log \boldsymbol{S}_i - \log \sum_i \boldsymbol{S}_i > 0. \tag{19}$$

Equivalently, we have

$$\boldsymbol{D}_i = -\log \boldsymbol{S}_i < 1 - \log \sum_i \boldsymbol{S}_i = 1 - \log \sum_i e^{-\boldsymbol{D}_i}. \tag{20}$$

Therefore, we only need to find a case such that:

$$\forall i, \quad \boldsymbol{D}_i = -\log \boldsymbol{S}_i \geq 1 - \log \sum_i \boldsymbol{S}_i. \tag{21}$$

As $\boldsymbol{D}_i \geq 0$, we only need:

$$\log \sum_i \boldsymbol{S}_i \geq 1, \quad \sum_i \boldsymbol{S}_i \geq e. \tag{22}$$

which can be easily satisfied when the codebook size is large or the codewords are close to each other. This completes the proof. $\qquad \square$

## D.2 PROOF OF PROPOSITION 5.2

Proposition 5.2: Suppose that $p(z)$ follows a uniform distribution. Given the same sample input $\boldsymbol{y}$ and parameters $\boldsymbol{\theta}$ of the generative model $\log p_{\boldsymbol{\theta}}(\boldsymbol{x}|\boldsymbol{B}\boldsymbol{y}^T)$, we have $\log p_{\arg\min_{\boldsymbol{\theta}} \mathcal{L}_{RVQ}}(\boldsymbol{x}|\boldsymbol{B}\boldsymbol{y}^T) < \log p_{\arg\min_{\boldsymbol{\theta}} \mathcal{L}_{VQ}}(\boldsymbol{x}|\boldsymbol{B}\boldsymbol{y}^T)$ .

*Proof.* Suppose that

$$\hat{\boldsymbol{\theta}}, \hat{\boldsymbol{B}} = \arg\min_{\boldsymbol{\theta}, \boldsymbol{B}} \mathcal{L}_{VQ} = \arg\min_{\boldsymbol{\theta}, \boldsymbol{B}} \mathcal{L}_{RVQ}. \tag{23}$$

It follows from (4) that

$$\hat{\boldsymbol{\theta}} = \arg\max_{\boldsymbol{\theta}} \log p_{\boldsymbol{\theta}}(\boldsymbol{x}|\boldsymbol{B}\boldsymbol{y}^T), \quad \text{and,} \quad I(\boldsymbol{x}) = \boldsymbol{B}\boldsymbol{y}^T. \tag{24}$$

For RVQ,

$$\mathcal{L}_{RVQ} = H(q(\boldsymbol{z}|\boldsymbol{x})) - \log p_{\boldsymbol{\theta}}(\boldsymbol{x}|\boldsymbol{B}\boldsymbol{y}^T), \quad \frac{\partial \mathcal{L}_{RVQ}}{\partial \boldsymbol{B}} = \frac{\partial H(q(\boldsymbol{z}|\boldsymbol{x}))}{\partial \boldsymbol{D}} \frac{\partial \boldsymbol{D}}{\partial \boldsymbol{B}}. \tag{25}$$

According to the proof of Proposition 5.1, we can obtain $\partial H(q(\boldsymbol{z}|\boldsymbol{x}))/\partial \boldsymbol{D} = 0$ only if

$$\forall i, \boldsymbol{D}_i = -\log \boldsymbol{S}_i = 1 - \log \sum_i \boldsymbol{S}_i. \tag{26}$$

In other words, $\boldsymbol{D}_1 = \boldsymbol{D}_2 = ... = 1 - \log \sum_i \boldsymbol{S}_i$, which means that $q(\boldsymbol{z}|\boldsymbol{x})$ is a uniform distribution. Given (24), the above equality is satisfied only when $\boldsymbol{B}_1 = \boldsymbol{B}_2 = ... = I(\boldsymbol{x})$, that is, the codebook collapses.

Apparently, this is not the global optimal point of the codebook for $\mathcal{L}_{RVQ}$, which contradicts with (23). Therefore,

$$\log p_{\arg\min_{\boldsymbol{\theta}, \boldsymbol{B}} \mathcal{L}_{RVQ}}(\boldsymbol{x}|\boldsymbol{B}\boldsymbol{y}^T) < \log p_{\hat{\boldsymbol{\theta}}, \hat{\boldsymbol{B}}}(\boldsymbol{x}|\boldsymbol{B}\boldsymbol{y}^T) = \log p_{\arg\min_{\boldsymbol{\theta}, \boldsymbol{B}} \mathcal{L}_{VQ}}(\boldsymbol{x}|\boldsymbol{B}\boldsymbol{y}^T). \tag{27}$$

This completes the proof. $\qquad \square$

## D.3 RELATIONSHIP TO SQ-VAE (TAKIDA ET AL., 2022)

SQ-VAE (Takida et al., 2022) introduces a stochastic quantization and dequantization process into the latent model of the relaxed VQ. The quantized latent variable is denoted as $\boldsymbol{z}_q$ and the dequantized latent variable as $\boldsymbol{z}$. The inference model can be represented as $q_1(\boldsymbol{z}|\boldsymbol{x})q_2(\boldsymbol{z}_q|\boldsymbol{z})$ and the generative model as $p(z_q)p_1(\boldsymbol{x}|\boldsymbol{z}_q)p_2(\boldsymbol{z}|\boldsymbol{z}_q)$. The optimization objective of SQ-VAE is the negative ELBO:

$$\mathcal{L}_{SQ} = \mathbb{E}_{q_1(\boldsymbol{z}|\boldsymbol{x})q_2(\boldsymbol{z}_q|\boldsymbol{z})} \log \frac{p_2(\boldsymbol{z}|\boldsymbol{z}_q)}{q_1(\boldsymbol{z}|\boldsymbol{x})} - \mathbb{E}_{q_2(\boldsymbol{z}_q|\boldsymbol{z})} \log q_2(\boldsymbol{z}_q|\boldsymbol{z}) - \mathbb{E}_{q_2(\boldsymbol{z}_q|\boldsymbol{z})} \log p(\boldsymbol{z}_q) - \log p(\boldsymbol{x}|\boldsymbol{z}_q). \tag{28}$$

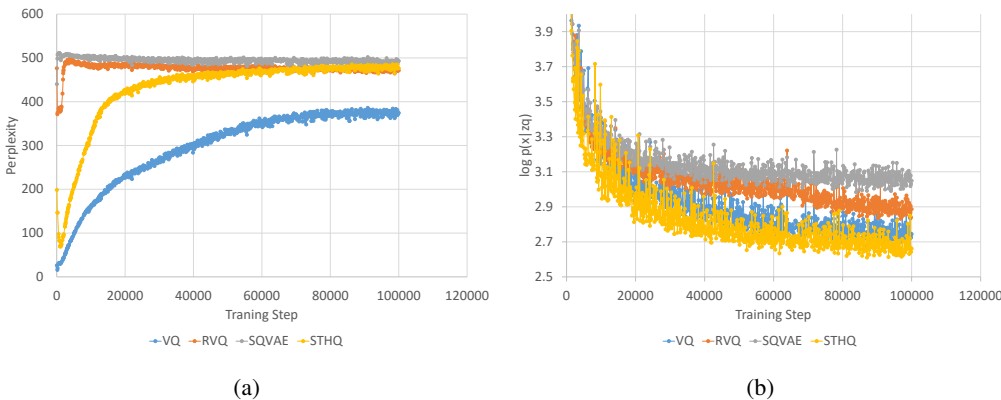

Figure 5: Convergence processes for different quantization methods.

Let us further assume that $z$ is a Gaussian distribution with the same variance $\sigma_\phi$ in both $p_2$ and $q_1$. Eq. (28) can be simplified as:

$$\mathcal{L}_{SQ} = \frac{||\boldsymbol{z} - \boldsymbol{z}_q||_2}{2\sigma_\phi^2} - \mathbb{E}_{q_2(\boldsymbol{z}_q|\boldsymbol{z})} \log q_2(\boldsymbol{z}_q|\boldsymbol{z}) - \mathbb{E}_{q_2(\boldsymbol{z}_q|\boldsymbol{z})} \log p(\boldsymbol{z}_q) - \log p(\boldsymbol{x}|\boldsymbol{z}_q). \quad (29)$$

Recall that the loss function of the proposed STHQ is

$$\mathcal{L}_{STHQ} = ||\boldsymbol{D}_{min}||_2 - \mathbb{E}_q \log p(\boldsymbol{z}) - \log p(\boldsymbol{x}|\boldsymbol{B}\boldsymbol{y}^T). \quad (30)$$

We can tell that SQ-VAE is equivalent to STHQ when $2\sigma_\phi^2 = 1$, the entropy term is removed, and the Gumbel-softmax sampling for the categorical distribution is replaced by the proposed $\mathrm{st\text{-}hardmax}$ sampling for $q_2(\boldsymbol{z}_q|\boldsymbol{z})$. As discussed in Section 2, SQ-VAE requires $\log p(\boldsymbol{x}|\boldsymbol{z}_q)$ to degenerate so as to ensure $\sigma_\phi \to 0$. However, for compression tasks, $\log p(\boldsymbol{x}|\boldsymbol{z}_q)$ needs to be non-degenerate. Consequently, $\frac{||\boldsymbol{z}-\boldsymbol{z}_q||_2}{2\sigma_\phi^2}$ approaches infinity if $\sigma_\phi \to 0$. In other words, the stochastic quantization in SQ-VAE is not applicable to our problem, and thus, we propose STHQ as a remedy.

### D.4 EMPIRICAL ANALYSIS OF DIFFERENT QUANTIZATION METHODS

In order to further substantiate the preceding discussions, we conducted optimization experiments employing different quantization methods, namely VQ, RVQ, SQ-VAE, and the proposed STHQ. The experimental setup used in Section 6.2 was adopted for training. Following the methodology outlined in (Takida et al., 2022) for SQ-VAE, we employed codebook perplexity as a measure of codebook utilization, and the likelihood ($\log p(\boldsymbol{x}|\boldsymbol{z}_q)$) as an indicator of the quality of reconstruction throughout the entire training process. The results are presented in Figure 5.

It is obvious that STHQ achieves the best reconstruction result and the codebook utilization is very close to RVQ and SQ-VAE. Analysis of the codebook perplexity results reveals that RVQ and SQ-VAE achieve a more representative codebook compared to VQ, owing to the global update property facilitated by the entropy term. However, their likelihood remains inferior to that of VQ, as confirmed by Proposition 5.2. In contrast, the proposed STHQ approach achieves a perplexity value comparable to that of RVQ and SQ-VAE, while exhibiting faster convergence of likelihood than VQ. This notable improvement can be attributed to the global update capability and the mitigation of the entropy gap problem enabled by STHQ, resulting in accelerated convergence of both the codebook and the likelihood.

## E EXTRA EXPERIMENTAL RESULTS

### E.1 FULL EXPERIMENT RESULTS FOR SECTION 6.2

Here we present the compression time of the backbone network and entropy coders for each method in Table 1. The results, displayed in Table 5, indicate that for most methods, the compression time

Table 5: Experimental results of latent coding Methods. Expands upon Table 1 by including compression and decompression times for each method on CF10.

| Methods | | CF10 Comp Time (ms) | | | CF10 Decomp Time (ms) | | |
| --- | --- | --- | --- | --- | --- | --- | --- |
| | | Net | Coder | Total | Net | Coder | Total |
| Continuous | bits-back (Townsend et al., 2019a) | 32.07 | 4.54 | 36.61 | 32.44 | 9.85 | 42.29 |
| Discrete | UQ (Ballé et al., 2018) | 0.98 | 1.53 | 2.51 | 1.60 | 1.54 | 3.15 |
| | VQ (van den Oord et al., 2017) | 0.98 | 0.61 | 1.59 | 1.57 | 0.88 | 2.45 |
| | McQuic (Zhu et al., 2022) | 1.03 | 11.25 | 12.28 | 1.57 | 8.75 | 10.32 |
| Autoregressive | MaskConv2D (van den Oord et al., 2016b) | 1.03 | 4.26 | 5.28 | 1.75 | 632.24 | 633.99 |
| | Checkerboard (He et al., 2021) | 1.03 | 4.65 | 5.67 | 1.63 | 4.71 | 6.34 |
| Proposed | tANS (Duda, 2013) | 0.99 | 0.92 | 1.91 | 1.66 | 0.86 | 2.52 |
| | FSAR(O1) | 0.98 | 1.00 | 1.98 | 1.58 | 0.97 | 2.55 |
| | FSAR(O2) | 0.99 | 1.08 | 2.07 | 1.62 | 1.04 | 2.65 |
| | FSAR(O2)+LSN | 0.98 | 1.47 | 2.45 | 1.59 | 1.06 | 2.65 |
| | FSAR(O3) | OOM | OOM | OOM | OOM | OOM | OOM |

Table 6: Experimental results of lossless image compressors on different datasets. BPD values for IDF and IDF++ on CF10, IN32, and IN64 datasets are obtained from their respective original papers.

| Methods | | Practical BPD | | | | | Speed (MB/s) | |
| --- | --- | --- | --- | --- | --- | --- | --- | --- |
| | | CF10 | IN32 | IN64 | Kodak | CLIC20 | CSpd | DSpd |
| Traditional | PNG | 5.78 | 6.09 | 5.42 | 4.53 | 4.23 | 16.559 | 59.207 |
| | WebP | 4.52 | 4.95 | 4.32 | 3.20 | 3.08 | 1.762 | 26.688 |
| | FLIF | 4.29 | 4.78 | 4.25 | 2.88 | 2.82 | 0.858 | 2.715 |
| Autoencoder | L3C | 4.77 | 4.89 | 4.29 | 3.55 | 3.34 | 0.036 | 0.057 |
| | bitswap | 3.84 | 4.54 | - | - | - | 0.003 | 0.004 |
| | PILC | 4.28 | 4.95 | 4.51 | 3.68 | 3.55 | 2.659 | 3.725 |
| Flow | IDF | 3.34 | 4.18 | 3.90 | - | - | 0.010 | 0.007 |
| | IDF++ | 3.26 | 4.12 | 3.81 | - | - | 0.009 | 0.007 |
| Proposed | PILC+STHQ | 4.29 | 4.90 | 4.50 | 3.68 | 3.57 | 2.566 | 3.795 |
| | PILC+STHQ+FSAR | 4.18 | 4.80 | 4.39 | 3.66 | 3.51 | 2.378 | 3.590 |

closely aligns with the decompression time. However, the MaskConv2D autoregressive method demonstrates a notable distinction, benefitting from its parallel computation during the compression process.

### E.2 FULL EXPERIMENT RESULTS FOR SECTION 6.3

We present comprehensive results in Figure 4, encompassing experiments conducted on ImageNet32/64, as well as a high-resolution benchmark utilizing the Kodak (Kod, 2023) dataset and CLIC 2020 (CLI, 2023) validation set (referred as to Kodak and CLIC20 respectively). The Kodak dataset consists of images with a resolution of $512 \times 768$, while the CLIC dataset ranges from $720 \times 439$ to $2048 \times 1370$ in resolution. It is important to note that our model is trained following the methods described in (Mentzer et al., 2018) and (Kang et al., 2022), using a subset of the preprocessed Open Images training dataset (Kuznetsova et al., 2018). Furthermore, our evaluation includes lossless codecs based on flow models, specifically integer discrete flows (IDF) (Hoogeboom et al., 2019) and its enhanced version IDF++ (van den Berg et al., 2020). The corresponding results are presented in Table 6.

The results of the PILC+STHQ+FSAR method on IN32/64 demonstrate similarity to CIFAR10, exhibiting similar BPD values to WebP and FLIF, and slightly inferior performance compared to PILC. In the case of high-resolution images, the PILC+STHQ+FSAR method performs marginally worse than WebP but outperforms PNG, likely due to the limited versatility of the shallow backbone. This implies the necessity for improved backbones when dealing with large-scale images. However, it is important to note that the proposed STHQ and FSAR techniques enhance the performance of PILC across all datasets, validating the effectiveness and robustness of the proposed method for lossless

Table 7: BPD results (with standard deviation) for using different autoregressive Markov models on different spaces. The standard deviations are presented in the brackets.

|          | Observation | Latent      |
|----------|-------------|-------------|
| None     | 8.00        | 5.23 (0.02) |
| MaskConv | 5.37(0.00)  | 5.05(0.04)  |
| FSAR(O1) | 6.19(0.00)  | 5.15(0.03)  |
| FSAR(O2) | 5.49(0.01)  | 5.06(0.02)  |
| FSAR(O3) | 5.26(0.01)  | 4.96(0.03)  |

image compression tasks. Regarding flow models, IDF and IDF++ surpass all other methods in terms of BPD, although their slower processing speed remains a critical concern.

### E.3 COMPARISON OF DIFFERENT USAGES OF AUTOREGRESSIVE MARKOV MODELS

As discussed in Section 4, the use of autoregressive Markov models as the latent prior is considered an optimal solution. This is because the latent space is typically smaller than the observation space, and the dependency between variables is relatively weak in the latent space. As a result, a more compact autoregressive Markov model, such as the FSAR model, can be employed. In this section, we empirically demonstrate the validity of this assertion by comparing the performance of various autoregressive Markov models. These models include the proposed FSAR model, a higher-order autoregressive Markov model (order-3 FSAR), and a Masked Convolution based model (van den Oord et al., 2016b). The comparison is conducted for both observation space and latent space scenarios using the CIFAR10 dataset. The theoretical BPD is reported for this experiment.

The results demonstrate that, in the observation space, more complex autoregressive Markov models generally yield better compression performance than their simpler counterparts. This suggests that the dependencies between variables are more intricate in the observation space than in the latent space. Conversely, applying higher-order Markov models does not result in significant improvements in the latent space. This highlights the advantages of employing compact, low-order autoregressive Markov models as the latent prior.

### E.4 COMPARISON OF DIFFERENT HYPERPARAMETERS FOR OPTIMIZING STHQ

This study aims to establish the superiority of the proposed STHQ method over existing VQ-based methods. To compare these models, various hyperparameters are considered, including backbone network settings, latent space, and codebook size. The experiments are conducted on the CIFAR10 dataset. Specifically, two backbone networks are evaluated: a shallow backbone (consisting of one downsampling or upsampling layer, and one residual layer with 64 latent channels) used in previous experiments, and a deeper backbone (consisting of two downsampling or upsampling layers, and two residual layers with 256 latent channels). For the shallow backbone, the latent variables are quantized to 1, 2, and 4 channels, with corresponding codeword channels of 64, 32, and 16, respectively. For the deeper backbone, the latent variables are quantized to 4, 8, and 16 channels, with corresponding codeword channels of 64, 32, and 16, respectively. The proposed STHQ method is compared to VQ-VAE (van den Oord et al., 2017) (denoted as VQ), Relaxed VQ-VAE (Sønderby, 2017) (denoted as RVQ) and SQ-VAE (Takida et al., 2022) (denoted as SQ). The results are summarized in Table 8.

Our findings reveal that the proposed STHQ method outperforms all other VQ-based methods in most configurations, except for a few cases where the compression performance of STHQ is comparable to other methods. Generally, VQ-VAE performs relatively worse when quantized to fewer channels (1 or 2), but better when quantized to more channels (4, 8, or 16). Conversely, the relaxed VQ-VAE and SQ-VAE exhibit relatively higher performance when quantized to fewer channels, but lower performance for more channels. In contrast, the proposed STHQ method consistently achieves comparable or superior results across all configurations, demonstrating its robustness in optimizing discrete latent space.

Table 8: BPD results (with standard deviation) for using different hyperparameters for different VQ based methods. The standard deviations are presented in the brackets.

| Hyperparameters | | | | Method | | | |
|---|---|---|---|---|---|---|---|
| Backbone | Latent | Quant | Codeword | VQ | RVQ | SQ | Proposed |
| 1-layer | 64 | 1 | 64 | 5.75 (0.04) | 5.60 (0.04) | **5.48 (0.01)** | **5.44 (0.02)** |
| | | 2 | 32 | 5.34 (0.07) | 5.81 (0.18) | 5.62 (0.22) | **5.26 (0.06)** |
| | | 4 | 16 | 5.87 (0.03) | 6.67 (0.02) | 6.91 (0.05) | **5.66 (0.08)** |
| 2-layers | 256 | 4 | 64 | **5.00 (0.05)** | 5.09 (0.10) | 5.35 (0.06) | **4.96 (0.03)** |
| | | 8 | 32 | 4.87 (0.02) | 5.23 (0.03) | 6.02 (0.02) | **4.79 (0.02)** |
| | | 16 | 16 | **5.20 (0.07)** | 6.30 (0.04) | 7.40 (0.15) | **5.17 (0.03)** |

Table 9: Generation task results for different VQ-based methods.

| | NLL | FID | Perplexity |
|---|---|---|---|
| VQ | 2.83 | 58.00 | 262.93 |
| RVQ | 2.88 | 64.15 | 327.68 |
| SQ | 2.96 | 83.09 | **489.07** |
| EMA-VQ | 2.80 | 48.59 | 380.45 |
| EM-VQ | 2.76 | 55.76 | 282.97 |
| Anneal-RVQ | 3.12 | 81.02 | 61.55 |
| Proposed | **2.67** | **42.63** | **487.75** |

## E.5 COMPARISON OF VQ-BASED METHODS ON GENERATION TASKS

Finally, we conduct a comparative analysis of our proposed STHQ method against existing vector quantization (VQ)-based approaches in simple image generation tasks. In addition to the methods listed in Table 8, we include comparisons with VQ variants, namely exponential moving average-based codebook update for VQ-VAE (EMA-VQ) (Polyak & Juditsky, 1992), expectation maximization optimizer for VQ-VAE (EM-VQ) (Roy et al., 2018), and relaxed VQ with exponential deterministic annealing (Anneal-RVQ) (Rose, 1998), where the softmax temperature is annealed from 1.0 and halved every 200 epochs. In our evaluation, we employ the negative log-likelihood (NLL) metric, commonly used for assessment, along with the Frechet Inception Distance (FID) (Heusel et al., 2017) to gauge the quality of the generated images. Additionally, we utilize perplexity as a measure of codebook utilization, where a higher perplexity indicates better codebook training. These experiments are conducted on CIFAR10, utilizing the same hyperparameters as described in Section 6.2. The results are presented in Table 9.

The results reveal that the proposed STHQ method outperforms all other approaches across all metrics. Among the VQ variants, EMA-VQ and EM-VQ demonstrate superior performance compared to traditional VQ, while Anneal-RVQ exhibits notably worse results, potentially due to an improper annealing schedule. This suggests that tuning deterministic annealing proves challenging in practical applications. On the other hand, the proposed STHQ method achieves better performance than both EMA and EM optimization techniques, while also avoiding the complexities associated with tuning the annealing process.

## F EXPERIMENT IMPLEMENTATION DETAILS

### F.1 HARDWARE SETUP

To ensure fair comparisons, we conduct all experiments under the same hardware configuration. For CPU-based experiments, we utilize a desktop machine equipped with an Intel i7-6800K CPU. For GPU-based model training, we employ a virtual machine on a Kubermaker cluster featuring 8 NVIDIA V100 GPUs.

### F.2 NETWORK STRUCTURE

In our experiments, unless otherwise specified, we maintain consistency in the backbone network across all methods to ensure that the compression ratio remains unaffected by variations in the backbone architecture. Our focus on general computation devices, such as CPUs, leads us to adopt a shallow backbone network based on ResNet (He et al., 2015). This selection aims to minimize the number of floating-point operations (FLOPs) involved. The backbone network comprises a single downsampling (or upsampling) convolutional layer and one residual block within the inference (or generative) model. Consequently, the inference model performs 2x downsampling, while the generative model performs 2x upsampling. With regard to the latent space, we set the number of input latent channels to 64 and quantize them into 2 channels, each containing 256 codewords. This configuration leads to a codebook size of $2 \times 256 \times 32$, where 2 represents the quantized latent channels, 256 denotes the number of codewords (also known as state number), and 32 signifies the codeword dimension. Consequently, a $3 \times 32 \times 32$ input image corresponds to $2 \times 16 \times 16$ latent variables.

### F.3 HYPERPARAMETERS AND CONFIGURATIONS OF THE PROPOSED METHODS

Regarding FSAR, each variable relies on its previous neighboring variables in the spatial domain. Specifically, the latent prior utilizing the Order-1 Markov model is defined as $p(y_{i,j,k}|y_{i,j,k-1})$, while Order-2 is $p(y_{i,j,k}|y_{i,j-1,k}, y_{i,j,k-1})$, and Order-3 is $p(y_{i,j,k}|y_{i,j-1,k-1}, y_{i,j-1,k}, y_{i,j,k-1})$, where $i, j, k$ represent the indices of the channel, height, and width dimensions, respectively. During training, the Markov models are implemented using a 3-layer network consisting of 3 Linear layers and 2 ReLU layers. For the learnable state number, the initial number of states is set to 256, and $\alpha = 1.5$ is used in the $\alpha$-entmax method. For STHQ, the temperature for Gumbel-softmax $\mathrm{GS}_\tau$ is set as a constant 0.5.

### F.4 TRAINING PROCEDURE

The training process utilized the Adam optimizer with a learning rate of $10^{-3}$. For the CIFAR10 dataset, the training was performed for 1000 epochs, with a batch size of 64 per GPU. Regarding the ImageNet32 dataset, the training was conducted for 50 epochs, with a batch size of 64 per GPU. As for ImageNet64, the training setup was identical to ImageNet32, except that the batch size was set to 16 per GPU.

It is important to note that in the STHQ-based experiments, the Gumbel-softmax reparameterization in the st-hardmax operator maintained a constant temperature throughout the training process. In this case, a temperature of 1.0 was employed as we required its gradient for optimization purposes. Additionally, the $\log p(\boldsymbol{x}|\boldsymbol{B}\boldsymbol{y}^T)$ term in the loss function (9) was implemented using a straight-through estimator, similar to VQ-VAE (van den Oord et al., 2017). This approach, rather than directly utilizing the samples in the proposed st-hardmax, proved to be beneficial for effectively optimizing the inference network.

### F.5 IMPLEMENTATION DETAILS FOR THE LATENT SPACE MODELS IN TABLE 1

Here we provide implementation details for latent space models used in Table 1. First, Table 10 illustrates the overall architecture of each method by linking them to Figure 1. Generally, for discrete latent space (Figure 1b), the variations among different methods primarily stem from the choice of quantization method. On the other hand, for models utilizing a discrete autoregressive latent space (Figure 1c), the main point of differentiation lies in the context model. Regarding our proposed FSAR-tANS model (Figure 1d), we utilize the STHQ method for quantization and conduct experiments using different FSAR models.

Next, we provide detailed explanations for the compared methods. Unless otherwise specified, all models employed in this study adhere to the same training scheme and utilize identical inference and generative networks. The reported values are obtained by averaging the corresponding metrics from three independent experiments. It is worth mentioning that the memory occupation values only take into account statically allocated data, such as the lookup table for tANS and the kernel parameters for MaskConv. In the case of LSN, only valid states are utilized to construct lookup tables. Therefore, the number of reduced states is consistent with the memory savings, as the lookup

Table 10: Architecture and Implementation Details of methods in Table 1.

| Architecture | Method | Detail |
|---|---|---|
| Figure 1a | bits-back (Townsend et al., 2019a) | - |
| Figure 1b | UQ (Ballé et al., 2018)
VQ (van den Oord et al., 2017)
McQuic (Zhu et al., 2022) | quant: uniform
quant: VQ
quant: 3-layer VQVAE |
| Figure 1c | MaskConv2D (van den Oord et al., 2016b)
Checkerboard (He et al., 2021) | context: MaskConv2D
context: Checkerboard |
| Figure 1d | tANS (Duda, 2013)
FSAR(O1)
FSAR(O2)
FSAR(O2)+LSN
FSAR(O3) | FSAR-tANS: non-autoregressive
FSAR-tANS: Order-1 Markov
FSAR-tANS: Order-2 Markov
FSAR-tANS: Order-2 Markov + LSN
FSAR-tANS: Order-3 Markov |

table takes $\mathcal{O}(C^N)$ space. For example, the FSAR(O2)+LSN ($C = 256, N = 2$) approach achieves an average of 187.33 states, which can be also inferred from the memory occupation saving from Table 1: $256(276.586/516.512)^{1/2} \approx 187.33$.

**STHQ+tANS (Duda, 2013)** For the STHQ+tANS method, we utilize the tANS implementation from the zstd library[1], in accordance with the conventions followed in other STHQ-based experiments such as FSAR and various autoregressive-based latent models.

**bits-back (Townsend et al., 2019a)** The bits-back method is implemented using the craystack library[2] provided by the authors.

**UQ (Ballé et al., 2018)** For universal quantization latent coding, we employ the entropy bottleneck proposed in (Ballé et al., 2018) and implemented in CompressAI[3]. The size of the latent channels is set to 64, which is equivalent to the output channels of the inference network.

**VQ (van den Oord et al., 2017)** The VQ model is implemented using the original implementation of VQ-VAE[4] (van den Oord et al., 2017), with tANS used for constructing the latent coder. The codebook size is the same as that used in the STHQ-based experiments, which is $2 \times 256 \times 32$.

**McQuic (Zhu et al., 2022)** We adhere to the official McQuic repository[5] for implementation, replacing the entropy coder with tANS for a fair speed comparison. We adopt the recommended hyperparameters $m = 2$ and $K = [8192, 2048, 512]$, and the size of the latent channels remains consistent with other experiments, which is 64.

**MaskConv2D (van den Oord et al., 2016b)** We employ a single masked convolution layer, using mask A as described in (van den Oord et al., 2016b), as an autoregressive model for latent variables. The entropy coder used is the same as in STHQ+tANS. During latent entropy coding, the autoregressive model is iterated along the spatial dimensions.

**Checkerboard (He et al., 2021)** Similar to MaskConv2D, we utilize a single masked convolution layer as the autoregressive model, adopting a checkerboard-form mask as described in (He et al., 2021). The latent entropy coding is iterated in two subsequent steps.

F.6 IMPLEMENTATION DETAILS FOR THE COMPRESSION METHODS IN TABLE 6

**PNG, WebP** To compress images in PNG and lossless WebP formats, we utilize the pillow library[6].

---

[1] https://github.com/facebook/zstd
[2] https://github.com/j-towns/craystack
[3] https://github.com/InterDigitalInc/CompressAI
[4] https://github.com/bshall/VectorQuantizedVAE
[5] https://github.com/xiaosu-zhu/McQuic
[6] https://pypi.org/project/Pillow/

**FLIF** For FLIF compression, we use the imageio-flif python library[7] which binds the official C code[8], and we execute "imageio.v2.imwrite(bytesio, image, format="FLIF", disable_color_buckets=True)" to compress images.

**L3C (Mentzer et al., 2018)** To test using the L3C method, we employ the official code[9] and the provided pretrained models for different datasets.

**bitswap (Kingma et al., 2019)** For bitswap compression, we use the official code[10] and the provided pretrained model (labeled "nz2") for testing on CIFAR10 and ImageNet32 (ImageNet64 is skipped as no pretrained models is provided). It should be noted that, unlike other experiments, all images are stored in the same bitstream, which is one reason for the method's low BPD.

**PILC (Kang et al., 2022)** Since the code for PILC is currently unavailable, we implement this method in our benchmark. This includes the three-way autoregressive (TWAR) model for image prediction and the AI-ANS coder for entropy coding, as described in (Kang et al., 2022). Specifically, we adopt the recommended backbone network from the authors, consisting of one downsampling (or upsampling) convolutional layer and four residual blocks in the inference (or generative) model. It is worth mentioning that the code for PILC is not publicly available, and therefore we have independently implemented this method in our benchmark. Our implementation includes the proposed three-way autoregressive (TWAR) model for image prediction and the AI-ANS coder (on CPU) for entropy coding. We use the same backbone network architecture as our experiments, which consists of one downsampling (or upsampling) convolutional layer and one residual block in the inference (or generative) model. The proposed method combined with PILC also follows this backbone architecture.

## G  DISCUSSIONS AND FUTURE WORKS

Despite the promising outcomes achieved through the proposed finite-state autoregressive entropy coding approach, there are still certain limitations that require attention. Notably, our experimental utilization of the lookup table still demands approximately 300MB of storage, even after reducing the number of states using the proposed Learnable State Number approach. This presents a significant hurdle for implementing the approach on portable devices and emphasizes the necessity for future research on maximizing the advantages of the autoregressive Markov model while keeping the space complexity limited.

One potential solution is to restrict the number of probability distributions employed for tANS coding, thereby limiting the corresponding state transition lookup tables used by tANS. By doing so, we only need to store pointers to these restricted distributions in the FSAR lookup table **L**. Assuming we limit the distributions to $D$ and each corresponding state transition lookup table contains $S$ states, for order-2 FSAR models, the minimal memory requirement is $DS\lceil\log_{256} S\rceil + C^2\lceil\log_{256} D\rceil$ bytes. In contrast, unlimited distributions would necessitate $C^2 S\lceil\log_{256} S\rceil$ bytes. In our experiments, setting $D = 256$ reduced memory usage from approximately 512MB to around 2MB. We leave determining the optimal selection of these limited distributions to achieve minimal performance degradation to future work.

Furthermore, it would be advantageous for future studies to consider adaptive selection of the Markov model structure. For instance, an intriguing approach has been explored in (Lin et al., 2023), where the authors attempt different decoding orders for parallel autoregressive models and select the optimal one during training. Similar methodologies could potentially be applied to the structure learning of FSAR Markov models.

Finally, it is also interesting to investigate the acceleration of the backbone network by means of methods such as neural architecture search. Indeed, Yang et al. (2021) have already delved into this direction in compression codecs using slimmable neural networks (Yu et al., 2018).

---

[7]https://codeberg.org/monilophyta/imageio-flif
[8]https://github.com/FLIF-hub/FLIF
[9]https://github.com/fab-jul/L3C-PyTorch
[10]https://github.com/fhkingma/bitswap

