# OpenReview forum: "Finite-State Autoregressive Entropy Coding for Efficient Learned Lossless Compression"
_ICLR.cc/2024/Conference — ICLR 2024 spotlight_

### Official Review · Reviewer_7HPB · 2023-10-17

**Soundness:** 3 good
**Presentation:** 2 fair
**Contribution:** 2 fair
**Rating:** 6
**Confidence:** 4

**Summary:**

This paper investigates efficient learnt lossless data compression, aiming to improve the compression performance while maintaining computational efficiency.
The model is based on autoencoder-based learnt codecs, with discrete autoregressive latent space.
The paper proposes Finite-State AutoRegressive (FSAR) entropy coder, which combines a low-complexity autoregressive Markov model with a fast entropy coder to achieve efficient latent coding.
The paper proposes Straight-Through Hardmax Quantization (STHQ) to opotimize the discrete latent space.
The proposed method improves the compression ratio by up to 6% compared to commonly used discrete deterministic autoencoders, with negligible additional computational time.

**Strengths:**

Most related works are properly cited and discussed, thus the proposed methods are well motivated.

The manuscript is well written and friendly to readers, with in-depth analysis of previous learnt compression architectures.

The experiments are thorough and extensive. I like the informative appendix.

Though the proposed FSAR and STHQ are improvement based on previous techniques as already explained in the manuscript, the technical novelty is enough from my point of view.

**Weaknesses:**

The memory consumption shown in Table 1 is significantly larger than previous methods.

It is great to compare the net and coder latency separately in Table 1, however, the coder of each methods are not explained. It is important to mark the coder of each methods since the latency of network is very close.

Only CIFAR10, ImageNet 32/64 are used in the experiments. It is better to show the scalability of the proposed methods on larger image sizes.

Missing citations. For efficient lossless image compression, I noticed there exist two very efficient design [R1, R2]. Although comparison is not required, these two paper should be discussed to better reflect recent progress regarding efficient learnt lossless image compression.
[R1] Guo, Lina, et al. "Practical Learned Lossless JPEG Recompression with Multi-Level Cross-Channel Entropy Model in the DCT Domain." Proceedings of the IEEE/CVF Conference on Computer Vision and Pattern Recognition. 2022.
[R2] Guo, Lina, et al. "Efficient Learned Lossless JPEG Recompression." arXiv preprint arXiv:2308.13287 (2023).

Minor:
In F.5 and F.6, the implementation details of checkerboard are not provided.

**Questions:**

It is not clear how the operation number in Table 4 is obtained, will this be included in the opensource code?

---

> ### Comment · Reviewer_7HPB · 2023-11-22
> **Any update?**
>
> Dear authors, any update according to those review comments?

---

> > ### Author Response · Authors · 2023-11-22
> > **Thank you for your patience and the response is on the way**
> >
> > Dear reviewer,
> >
> > Thank you for your patience. We are actively collecting and analyzing experimental results to address your comments. We will soon provide a revised version of our paper, along with a comprehensive response to all reviewer comments. Rest assured, we are mindful of the deadline and will submit our revised work on time. Your concern is greatly appreciated.

---

> ### Author Response · Authors · 2023-11-23
> **Response to Reviewer 7HPB (Part 1)**
>
> We sincerely thank you for your valuable comments and suggestions! We have gone through all your comments and questions, and accordingly conducted additional experiments, provided extra details, and clarified some questions in the revised paper. Below we address each comment or questions individually. The reviewer's comments are shown in italics. The paragraph(s) following them is the authors' response. The quoted paragraphs from our paper are labeled with a grey line left to the paragraphs. Unless specified, all references to pages, equations, and sections refer to the revised paper. Moreover, in the revised version of our paper, we mark all newly added or changed paragraphs in blue. For newly added or changed figures or tables, we mark their titles in blue instead. We will incorporate the suggested revisions into the final camera-ready version to enhance the clarity and persuasiveness of our paper.
>
> _Q1 - Missing citations. For efficient lossless image compression, I noticed there exist two very efficient design [R1, R2]. Although comparison is not required, these two paper should be discussed to better reflect recent progress regarding efficient learnt lossless image compression. [R1] Guo, Lina, et al. "Practical Learned Lossless JPEG Recompression with Multi-Level Cross-Channel Entropy Model in the DCT Domain." Proceedings of the IEEE/CVF Conference on Computer Vision and Pattern Recognition. 2022. [R2] Guo, Lina, et al. "Efficient Learned Lossless JPEG Recompression." arXiv preprint arXiv:2308.13287 (2023)._
>
> Reply: Thank you for bringing these two recent works to our attention. We have taken note of the relevance of these papers, as **they both incorporate parallelizable autoregressive models similar to the checkerboard method**. To reflect recent progress regarding efficient learnt lossless image compression, we have included a discussion of these works in the "Discrete Autoregressive Latent Space" paragraph within Section 2 on Page 3 of our paper.
> > Although recent works (Ruan et al., 2021; Ryder et al., 2022; Guo et al., 2022; 2023) develop parallelizable autoregressive models for efficient coding on parallel computation devices like GPUs...
>
>
> _Q2 - In F.5 and F.6, the implementation details of checkerboard are not provided._
>
> Reply: Thank you for the suggestions. We have added implementation details of autoregressive-based methods in Appendix F.5 on Page 24.
> > **MaskConv2D (van den Oord et al., 2016b)** We employ a single masked convolution layer, using mask A as described in (van den Oord et al., 2016b), as an autoregressive model for latent variables. The entropy coder used is the same as in STHQ+tANS. During latent entropy coding, the autoregressive model is iterated along the spatial dimensions.
>
> > **Checkerboard (He et al., 2021)** Similar to MaskConv2D, we utilize a single masked convolution layer as the autoregressive model, adopting a checkerboard-form mask as described in (He et al., 2021). The latent entropy coding is iterated in two subsequent steps.

---

> ### Author Response · Authors · 2023-11-23
> **Response to Reviewer 7HPB (Part 2)**
>
> _Q3 - Only CIFAR10, ImageNet 32/64 are used in the experiments. It is better to show the scalability of the proposed methods on larger image sizes._
>
> Reply: Thank you for your suggestion! In order to evaluate our method's performance on larger scale images, we have followed the approach taken by L3C and PILC. Specifically, we trained a checkpoint using a subset of the Open Images Dataset and conducted lossless compression tests on the Kodak and CLIC20 validation sets. However, it is important to note that due to limitations in terms of time and computational resources, we were only able to train on a smaller subset of approximately 100k images. This subset is smaller than that used by PILC, which employed around 400k images according to the L3C official code. Consequently, the resulting BPD values may be comparatively higher than usual.
>
> The results of these experiments can be found in Table 6, which is included in Appendix E.2 on Page 20 of the revised version of our paper. To provide further clarity, we would like to quote the relevant information from Appendix E.2 as follows:
>
> |  |  | Practical BPD |  |  |  |  | Speed (MB/s) |  |
> | --- | --- | --- | --- | --- | --- | --- | --- | --- |
> |  |  | CF10 | IN32 | IN64 | Kodak | CLIC20 | CSpd | DSpd |
> | Traditional | PNG | 5.78  | 6.09  | 5.42  | 4.53  | 4.23  | 16.559  | 59.207  |
> |  | WebP | 4.52  | 4.95  | 4.32  | 3.20  | 3.08  | 1.762  | 26.688  |
> |  | FLIF | 4.29  | 4.78  | 4.25  | 2.88  | 2.82  | 0.858  | 2.715  |
> | Autoencoder | L3C | 4.77  | 4.89  | 4.29  | 3.55  | 3.34  | 0.036  | 0.057  |
> |  | bitswap | 3.84  | 4.54  | - | - | - | 0.003  | 0.004  |
> |  | PILC | 4.28  | 4.95  | 4.51  | 3.86  | 3.79  | 2.659  | 3.725  |
> | Flow | IDF | 3.34 | 4.18  | 3.90  | - | - | 0.010  | 0.007  |
> |  | IDF++ | 3.26  | 4.12  | 3.81  | - | - | 0.009  | 0.007  |
> | Proposed | PILC+STHQ | 4.29  | 4.90  | 4.50  | 3.80  | 3.72  | 2.566  | 3.795  |
> |  | PILC+STHQ+FSAR | 4.18  | 4.80  | 4.39  | 3.74  | 3.65  | 2.378  | 3.590  |
>
> > We present comprehensive results in Figure 4, encompassing experiments conducted on ImageNet32/64, as well as a high-resolution benchmark utilizing the Kodak (Kod, 2023) and CLIC20 (CLI, 2023) datasets (referred as to Kodak and CLIC respectively).  The Kodak dataset consists of images with a resolution of $512\times 768$, while the CLIC dataset ranges from $720\times 439$ to $2048\times 1370$ in resolution. It is important to note that our model is trained following the methods described in (Mentzer et al., 2018) and (Kang et al., 2022), using a subset of the preprocessed Open Images training dataset (Kuznetsova et al., 2018) consisting of approximately 90,000 images. Furthermore, our evaluation includes lossless codecs based on flow models, specifically integer discrete flows (IDF) (Hoogeboom et al., 2019) and its enhanced version IDF++ (van den Berg et al., 2020). The corresponding results are presented in Table 6.
> > The results of the PILC+STHQ+FSAR method on IN32/64 demonstrate similarity to CIFAR10, exhibiting similar BPD values to WebP and FLIF, and slightly inferior performance compared to PILC. **In the case of high-resolution images, the PILC+STHQ+FSAR method performs marginally worse than WebP but outperforms PNG, likely due to the limited versatility of the shallow backbone. This implies the necessity for improved backbones when dealing with large-scale images. However, it is important to note that the proposed STHQ and FSAR techniques enhance the performance of PILC across all datasets, validating the effectiveness and robustness of the proposed method for lossless image compression tasks.** Regarding flow models, IDF and IDF++ surpass all other methods in terms of BPD, although their slower processing speed remains a critical concern.
>
>
>
> _Q4 - It is not clear how the operation number in Table 4 is obtained, will this be included in the opensource code?_
>
> Reply: Thank you for pointing this out. The code may not clearly reflect the operation number. Instead, we've included the formula or pseudo-code of each method in Appendix C.1 in the revised version, which may explain the number of operations. **We mark all particular operations in colored boxes so that reader could clearly obtain the number of operations of each method. We refer readers to Table 3 and 4 on Page 15-16 in the revised paper.**

---

> ### Author Response · Authors · 2023-11-23
> **Response to Reviewer 7HPB (Part 3)**
>
> _Q5 - The memory consumption shown in Table 1 is significantly larger than previous methods._
>
> Reply: We have also recognized this limitation and discussed it in Appendix G on Page 25 of our paper. The discussion is presented as follows:
> > Despite the promising outcomes achieved through the proposed finite-state autoregressive entropy coding approach, there are still certain limitations that require attention. Notably, our experimental utilization of the lookup table still demands approximately 300MB of storage, even after reducing the number of states using the proposed Learnable State Number approach. This presents a significant hurdle for implementing the approach on portable devices and emphasizes the necessity for future research on maximizing the advantages of the autoregressive Markov model while keeping the space complexity limited.
> > One potential solution is to restrict the number of probability distributions employed for tANS coding, thereby limiting the corresponding state transition lookup tables used by tANS. By doing so, we only need to store pointers to these restricted distributions in the FSAR lookup table L. Assuming we limit the distributions to $D$ and each corresponding state transition lookup table contains $S$ states, for order-2 FSAR models, the minimal memory requirement is $ DS \lceil \log_{256}S \rceil + C^{2} \lceil \log_{256}D \rceil$ bytes. In contrast, unlimited distributions would necessitate $ C^{2} S \lceil \log_{256}S \rceil $ bytes. In our experiments, setting $D=256$ reduced memory usage from approximately 512MB to around 2MB. We leave determining the optimal selection of these limited distributions to achieve minimal performance degradation to future work.
>
>
>
> _Q6 - ...however, the coder of each methods are not explained. It is important to mark the coder of each methods since the latency of network is very close._
>
> Reply: In the revised version of our paper, we have linked each group of methods (Continuous, Discrete, Autoregressive, and Proposed) to their respective figures (Figure 1a to 1d) in Section 1, aiming to improve the coherence and understanding. Additionally, we have included an additional table in Appendix F.5 on Page 24 that specifically addresses the coding methods employed as follows:
>
> | Architecture | Method | Detail |
> | --- | --- | --- |
> | Figure 1a | bits-back | - |
> | Figure 1b | UQ | quant: uniform |
> |  | VQ | quant: VQ |
> |  | McQuic | quant: 3-layer VQVAE |
> | Figure 1c | MaskConv2D | context: MaskConv2D |
> |  | Checkerboard | context: checkerboard |
> | Figure 1d | tANS | FSAR-tANS: non-autoregressive |
> |  | FSAR(O1) | FSAR-tANS: Order-1 Markov |
> |  | FSAR(O2) | FSAR-tANS: Order-2 Markov |
> |  | FSAR(O2)+LSN | FSAR-tANS: Order-2 Markov + LSN |
> |  | FSAR(O3) | FSAR-tANS: Order-3 Markov |

---

> > ### Comment · Reviewer_7HPB · 2023-11-23
> >
> > Thanks for the reply.

---

### Official Review · Reviewer_DfKd · 2023-10-25

**Soundness:** 3 good
**Presentation:** 3 good
**Contribution:** 3 good
**Rating:** 6
**Confidence:** 4

**Summary:**

## Summary
* The authors propose a accelerated auto-regressive entropy coding system for discrete autoregressive coding. It utilizes a look-up-table (LUT) that somewhat reminds me of n-grams markov model. Further, it proposes a hard-max version of gumbel-softmax VQ, with the sampling step replaced by an arg-max operation. The empirical results support their claims well.

**Strengths:**

## Strength
* The finite state MC model seems to be a very interesting middle point between vanilla neural network based autoregressive model and vanilla n-gram model. It achieves a good balance between compression ratio and speed.
* The STHQ training of VQ-VAE looks interesting and experimental results show that the improvement of nll (bpd) is quite consistent over many hyperparameter setting. The proposed approach and result is interesting to general readers beyond compression community, as VQ-VAE are widely used in generative modeling (Latent-Diffusion, Dalle).

**Weaknesses:**

## Weakness
* I think the FSAR that authors propose is an entropy model, not entropy coder. Usually I only regard general coders such as AC, ANS, RANS / RANGE as entropy coder. And I tend to agree that autoregressive model, FSAR are only entropy model. Though they can be deeply coupled with entropy coder, I still think they are only entropy model.
* As the authors have setup VQ-VAEs with various architectures, I suggest the authors test more variant of VQ-VAE training, such as expoential moving average codebook, and expectation maximization VAE, beyond the current original VQ, gumbel-softmax VQ and SQ. Furthermore, subtle variants of gumbel-softmax VQ, e.g., soft-gumbel / ST-gumbel, penalize KL-term or not can also be studied. Furthermore, it is also interesting to evaluate the sample quality of those VQs, in terms of FID / IS. As the models with better nll (bpd) might not have better sample quality, and vice versa [A note on the evaluation of generative models]. This can provide a more comprehensive understanding of the proposed hardmax-gumbel training strategy. And make this paper more interesting to density estimation community.

**Questions:**

## Questions
* The FSAR seems to be a general approach that is not necessarily used with rANS. Can it be used with any FIFO coding approach?
* The hardmax gumbel VQ seems to converge to softmax gumbel VQ as temperature cools down. The current form of hardmax gumbel VQ can be viewed as a softmax VQ with forward temperature 0. Is that possible to tune the annealing of softmax gumbel VQ to achieve the same effect of hardmax gumbel VQ?

---

> ### Author Response · Authors · 2023-11-23
> **Response to Reviewer DfKd (Part 1)**
>
> We sincerely thank you for your valuable comments and suggestions! We have gone through all your comments and questions, and accordingly conducted additional experiments, provided extra details, and clarified some questions in the revised paper. Below we address each comment or questions individually. The reviewer's comments are shown in italics. The paragraph(s) following them is the authors' response. The quoted paragraphs from our paper are labeled with a grey line left to the paragraphs. Unless specified, all references to pages, equations, and sections refer to the revised paper. Moreover, in the revised version of our paper, we mark all newly added or changed paragraphs in blue. For newly added or changed figures or tables, we mark their titles in blue instead. We will incorporate the suggested revisions into the final camera-ready version to enhance the clarity and persuasiveness of our paper.
>
> _Q1 - As the authors have setup VQ-VAEs with various architectures, I suggest the authors test more variant of VQ-VAE training, such as expoential moving average codebook, and expectation maximization VAE, beyond the current original VQ, gumbel-softmax VQ and SQ. Furthermore, subtle variants of gumbel-softmax VQ, e.g., soft-gumbel / ST-gumbel, penalize KL-term or not can also be studied._
> _Furthermore, it is also interesting to evaluate the sample quality of those VQs, in terms of FID / IS. As the models with better nll (bpd) might not have better sample quality, and vice versa [A note on the evaluation of generative models]. This can provide a more comprehensive understanding of the proposed hardmax-gumbel training strategy. And make this paper more interesting to density estimation community._
>
> Reply: Thank you for providing insightful suggestions. We appreciate your input. In response, we have conducted a new ablation study that compares various VQ-based methods on generation tasks. This study can be found in Appendix E.5 on Page 22 of the revised paper. It includes the evaluation of additional VQ variants and reports on a wider range of metrics. Specifically, we quote the relevant results and discussions from Appendix E.5 as follows:
>
> |  | NLL | FID | Perplexity |
> | --- | --- | --- | --- |
> | VQ | 2.83  | 58.00  | 262.93  |
> | RVQ | 2.88  | 64.15  | 327.68  |
> | SQ | 2.96  | 83.09  | 489.07  |
> | EMA-VQ | 2.80  | 48.59  | 380.45  |
> | Anneal-RVQ | 3.12  | 81.02  | 61.55  |
> | EM-VQ | 2.76  | 55.76  | 282.97  |
> | Proposed | 2.67  | 42.63  | 487.75  |
>
> > Finally, we conduct a comparative analysis of our proposed STHQ method against existing vector quantization (VQ)-based approaches in simple image generation tasks. In addition to the methods listed in Table 8, we include comparisons with VQ variants, namely exponential moving average-based codebook update for VQ-VAE (EMA-VQ) (Polyak & Juditsky, 1992), expectation maximization optimizer for VQ-VAE (EM-VQ) (Roy et al., 2018), and relaxed VQ with exponential deterministic annealing (Anneal-RVQ) (Rose, 1998), where the softmax temperature is annealed from 1.0 and halved every 200 epochs. In our evaluation, we employ the negative log-likelihood (NLL) metric, commonly used for assessment, along with the Frechet Inception Distance (FID) (Heusel et al., 2017) to gauge the quality of the generated images. Additionally, we utilize perplexity as a measure of codebook utilization, where a higher perplexity indicates better codebook training. These experiments are conducted on CIFAR10, utilizing the same hyperparameters as described in Section 6.2. The results are presented in Table 9.
>
> > The results reveal that **the proposed STHQ method outperforms all other approaches across all metrics**. Among the VQ variants, EMA-VQ and EM-VQ demonstrate superior performance compared to traditional VQ, while Anneal-RVQ exhibits notably worse results, potentially due to an improper annealing schedule. This suggests that tuning deterministic annealing proves challenging in practical applications. On the other hand, the proposed STHQ method achieves better performance than both EMA and EM optimization techniques, while also avoiding the complexities associated with tuning the annealing process.

---

> ### Author Response · Authors · 2023-11-23
> **Response to Reviewer DfKd (Part 2)**
>
> _Q2 - I think the FSAR that authors propose is an entropy model, not entropy coder. Usually I only regard general coders such as AC, ANS, RANS / RANGE as entropy coder. And I tend to agree that autoregressive model, FSAR are only entropy model. Though they can be deeply coupled with entropy coder, I still think they are only entropy model._
>
> _The FSAR seems to be a general approach that is not necessarily used with rANS. Can it be used with any FIFO coding approach?_
>
> Reply: Thank you for pointing this out. We acknowledge that we may have misused the term "entropy coder" when referring to the combination of FSAR and tANS. It would be more accurate to describe FSAR on its own as an "entropy model." However, it is worth noting that **FSAR can indeed be utilized in conjunction with any entropy coder**.
>
> While the FSAR approach can be combined with different entropy coders, **our choice to use tANS in combination with FSAR is driven by the underlying similarity in their implementation, particularly in using lookup tables for accelerated processing**. This combination results in an efficient and effective entropy coding framework. Hence, we refer to the combination of FSAR and tANS as a complete "entropy coder."
>
> To address this concern, we have included a discussion in the revised version of our paper in Appendix C.2 "Implementation of FSAR-tANS" on Page 16 as:
> > In theory, FSAR has the potential to be combined with any entropy coders. However, for practical implementation efficiency, we have opted for tANS (Duda, 2013), which leverages lookup tables for accelerated processing. The algorithm for finite-state autoregressive entropy coding based on tANS, including the initialization, encoding, and decoding processes, is illustrated in Algorithm 1...
>
>
> _Q3 - The hardmax gumbel VQ seems to converge to softmax gumbel VQ as temperature cools down. The current form of hardmax gumbel VQ can be viewed as a softmax VQ with forward temperature 0. Is that possible to tune the annealing of softmax gumbel VQ to achieve the same effect of hardmax gumbel VQ?_
>
> Reply: Thank you for raising this interesting point. It is true that the hardmax Gumbel Vector Quantization (VQ) converges to softmax Gumbel VQ as the softmax temperature decreases. In fact, the current form of hardmax Gumbel VQ can be seen as a special case of softmax VQ with a forward temperature of 0.
>
> Regarding the possibility of achieving the same effect of hardmax Gumbel VQ through annealing in softmax Gumbel VQ, it is indeed possible to tune the annealing schedule. Annealing has been commonly employed to optimize quantizers and has also been utilized in Soft-to-Hard VQ (Agustsson, 2017). However, determining the appropriate annealing schedule can be a challenging task.
>
> In our research, we conducted an experiment using exponential deterministic annealing on the softmax temperature of VQ. However, the obtained negative log-likelihood (NLL) results were inferior to those of VQ under the same hyperparameters. This observation is presented below in Table 9 (also see Appendix E.5 on Page 22 of the revised paper). As a result, **we can conclude that the proposed Soft-to-Hard Quantization (STHQ) offers a more robust solution compared to the use of annealing alone**.
>
> |  | NLL | FID | Perplexity |
> | --- | --- | --- | --- |
> | VQ | 2.83  | 58.00  | 262.93  |
> | RVQ | 2.88  | 64.15  | 327.68  |
> | SQ | 2.96  | 83.09  | 489.07  |
> | EMA-VQ | 2.80  | 48.59  | 380.45  |
> | **Anneal-RVQ** | **3.12** | 81.02  | 61.55  |
> | EM-VQ | 2.76  | 55.76  | 282.97  |
> | Proposed | 2.67  | 42.63  | 487.75  |

---

### Official Review · Reviewer_QqM2 · 2023-10-31

**Soundness:** 3 good
**Presentation:** 2 fair
**Contribution:** 3 good
**Rating:** 6
**Confidence:** 4

**Summary:**

This paper studies lossless image compression and proposes a finite-state autoregressive model to model the data, relying on a lookup table to predict likelihoods. They also present a soft relaxation of VQ to improve optimization.

**Strengths:**

The paper focuses on improving compression rates without increasing computational needs. In Table 2, we see that the authors succeed at this, improving over PLIC without significantly slowing the method down.

The soft VQ variatn (STHQ) is sensible.

The idea to parameterize a lookup table with an MLP and rely on the discrete nature of the inputs is smart.

**Weaknesses:**

Presentation of Results
- I had a very hard time following the results. Some suggestions: Please clarify what architecture was used for Table 1. I did not find it in the text and initially was confused why the IN32 results are so much higher than what is typical in the literature
- Can you consider replacing Table 2 with a three figures like in PLIC? Where we have one figure for each dataset, and see DSpd vs. BPD? This would make it much much easier to understand the contribution of this paper. Also, some of the methods that have been published for a long time are missing in the table (IDF, IDF++), and I don't know where the 0.057 MB/s for L3C is coming from, interpolating from Table 54 in their paper, I arrive at 0.66MB/s (I was confused because I remember they were only 3x slower than FLIF)

Related work
In the context of STVQ, I am missing a citation to Agustsson et al, 2017, "Soft-to-Hard Vector Quantization for End-to-End Learning Compressible Representations", https://arxiv.org/abs/1704.00648. IIUC, they applied a similar Gumbal Softmax trick, but also used it in the forward pass (whereas STVQ uses argmax for the forward). Still, this seems related.

Minor
- Please remove the negative vspace around tables, it's very hard to know what is table and what is text.

**Questions:**

What is the state number you use? Ie if you use LSN, what is predicted?

---

> ### Author Response · Authors · 2023-11-23
> **Response to Reviewer QqM2 (Part 1)**
>
> We sincerely thank you for your valuable comments and suggestions! We have gone through all your comments and questions, and accordingly conducted additional experiments, provided extra details, and clarified some questions in the revised paper. Below we address each comment or questions individually. The reviewer's comments are shown in italics. The paragraph(s) following them is the authors' response. The quoted paragraphs from our paper are labeled with a grey line left to the paragraphs. Unless specified, all references to pages, equations, and sections refer to the revised paper. Moreover, in the revised version of our paper, we mark all newly added or changed paragraphs in blue. For newly added or changed figures or tables, we mark their titles in blue instead. We will incorporate the suggested revisions into the final camera-ready version to enhance the clarity and persuasiveness of our paper.
>
> _Q1 - Can you consider replacing Table 2 with a three figures like in PLIC? Where we have one figure for each dataset, and see DSpd vs. BPD? This would make it much much easier to understand the contribution of this paper._
>
> Reply: Thanks a lot for pointing this out! In response to your suggestion, we have made updates to Table 2 in the revised version of our paper. Instead of a table, it now consists of two subfigures presenting BPD versus CSpd and BPD versus DSpd plots specifically for CIFAR10. We hope this modification will provide a clearer visualization of the contribution of our paper. Additionally, we have relocated the table containing the full results, which includes the new compared methods and datasets, to Appendix E.2 on Page 20 for reference.
>
> _Q2 - Related work In the context of STVQ, I am missing a citation to Agustsson et al, 2017, "Soft-to-Hard Vector Quantization for End-to-End Learning Compressible Representations", _[_https://arxiv.org/abs/1704.00648._](https://arxiv.org/abs/1704.00648.)_ IIUC, they applied a similar Gumbal Softmax trick, but also used it in the forward pass (whereas STVQ uses argmax for the forward). Still, this seems related._
>
> Reply: Thank you for bringing up this work. The work you mentioned, Agustsson et al. (2017), introduces soft VQ, which utilizes normalized Euclidean distances as mixing weights for the codebook, instead of employing Gumbel softmax for sampling. This can be confirmed from the "Soft assignments" paragraph in Section 3.2 of their paper ([https://arxiv.org/pdf/1704.00648.pdf](https://arxiv.org/pdf/1704.00648.pdf)), that is,
> >  Using soft assignment, we define the soft quantization of $\bar{z}$ as:
> > $\tilde{Q}(\bar{z}):=\mathbf{C}\phi(\bar{z})$
>
> In our paper, we have analyzed this method in the "Discrete Deterministic Latent Space" paragraph within Section 2 on Page 3, as follows:
> > **Instead, Agustsson et al. (2017) introduce soft VQ with deterministic annealing to enable optimization through hard quantization, but the practical implementation of the annealing schedule requires careful tuning.** Moreover, Takida et al. (2022) propose self-annealed stochastic quantization (SQ-VAE) to make the stochastic VQ-VAE converge to deterministic quantization. Unfortunately, the convergence from SQ to VQ relies on the collapse of the likelihood term (i.e., the generative model) (Takida et al., 2022), which is impractical for lossless compression as the likelihood is required for entropy coding.
>
>
> _Q3 - Please remove the negative vspace around tables, it's very hard to know what is table and what is text._
>
> Reply: We have made adjustments to the layout to ensure clearer separation between the tables and surrounding text.
>
> _Q4 - Also, some of the methods that have been published for a long time are missing in the table (IDF, IDF++)_
>
> Reply: We have included IDF and IDF++ in the compression experiments, and their results can be found in **Table 6, which is included in Appendix E.2 on Page 20** of the revised version of our paper. Please note that due to time constraints, we were unable to train the models until convergence. Therefore, the resulting BPD values from the checkpoints may be unreliable. Instead, we have directly cited the BPD values from the papers of IDF and IDF++. These values are included in Table 6. However, we want to point out that the speed results obtained from our checkpoints remain valid and are reported in Table 6. For the implementation of IDF, we used the official code, while for IDF++, we implemented it based on the IDF method. The relevant part of Table 6 is listed below:
> |  |  | Practical BPD |  |  | Speed (MB/s) |  |
> | --- | --- | --- | --- | --- | --- | --- |
> |  |  | CF10 | IN32 | IN64 |  CSpd | DSpd |
> | Flow | IDF | 3.34 | 4.18  | 3.90  | 0.010  | 0.007  |
> |  | IDF++ | 3.26  | 4.12  | 3.81  | 0.009  | 0.007  |
>
> We discuss the results in Appendix E.2 on Page 20 as follows:
> > Regarding flow models, IDF and IDF++ surpass all other methods in terms of BPD, although their slower processing speed remains a critical concern.

---

> ### Author Response · Authors · 2023-11-23
> **Response to Reviewer QqM2 (Part 2)**
>
> _Q5 - ...and I don't know where the 0.057 MB/s for L3C is coming from, interpolating from Table 54 in their paper, I arrive at 0.66MB/s (I was confused because I remember they were only 3x slower than FLIF)_
>
> Reply: Thank you for bringing up this concern. It is important to clarify that **the measured time or speed in our evaluation was obtained by running our method on CPUs, as our approach is specifically designed for general-purpose computation devices.**
> The results provided in the original paper you mentioned may have been obtained using GPUs, as they utilize a deeper backbone architecture that performs slower on CPUs.
>
> To address this, we have made a clarification in Section 6 on Page 7 of the revised version of our paper  as follows:
> > We evaluated performance using four criteria: Bits Per Dimension (BPD) for compression ratio, compression speed (CSpd) and decompression speed (DSpd) measured in megabytes per second (MB/s) to assess time complexity, and occupied memory (Mem) in megabytes (MB) to assess space complexity. The measured time or speed was obtained running on CPUs, as our method targets general-purpose computation devices.
>
>
> _Q6 - What is the state number you use? Ie if you use LSN, what is predicted?_
>
> Reply: In Appendix F.2 and F.3, we provide an introduction to the codebook used in our method, which has dimensions of 2x256x32. Here, the number 2 represents the quantized latent channels, **256 denotes the number of codewords (also known as state number)**, and 32 signifies the codeword dimension. We have made this clarification in Appendix F.2 on Page 23 of the revised version of our paper with the following statement:
> >  With regard to the latent space, we set the number of input latent channels to 64 and quantize them into 2 channels, each containing 256 codewords. This configuration leads to a codebook size of $2\times256\times32$, where 2 represents the quantized latent channels, 256 denotes the number of codewords (also known as state number), and 32 signifies the codeword dimension. Consequently, a $3\times 32 \times 32$ input image corresponds to $2\times 16 \times 16$ latent variables.
>
> Additionally, we incorporate the concept of Learnable State Number (LSN), which can reduce the state number. **In Table 1, the FSAR(O2)+LSN approach achieves an average of 187.33 states, as reported based on the average of three independent experiments**. This information is clarified in Appendix F.5 on Page 23 of the revised paper. We state:
> > The reported values are obtained by averaging the corresponding metrics from three independent experiments. It is worth mentioning that the memory occupation values only take into account statically allocated data, such as the lookup table for tANS and the kernel parameters for MaskConv. In the case of LSN, only valid states are utilized to construct lookup tables. Therefore, the number of reduced states is consistent with the memory savings, as the lookup table takes $\mathcal{O}(C^{N})$ space. For example, the FSAR(O2)+LSN ($C=256, N=2$) approach achieves an average of 187.33 states, which can be also inferred from the memory occupation saving from Table 1: $256(276.586 / 516.512)^{1/2} \approx 187.33$.
>
>
> _Q7 - Please clarify what architecture was used for Table 1._
>
> Reply: Thank you for your advice. In the original version of our paper, we attempted to provide clarification by linking each group of methods (Continuous, Discrete, Autoregressive, and Proposed) to their corresponding figures (Figures 1a, 1b, 1c, and 1d) in Section 1. In the revised version, we have included an additional table in Appendix F.5 on Page 24 that specifically addresses this aspect, that is,
>
> | Architecture | Method | Detail |
> | --- | --- | --- |
> | Figure 1a | bits-back | - |
> | Figure 1b | UQ | quant: uniform |
> |  | VQ | quant: VQ |
> |  | McQuic | quant: 3-layer VQVAE |
> | Figure 1c | MaskConv2D | context: MaskConv2D |
> |  | Checkerboard | context: Checkerboard |
> | Figure 1d | tANS | FSAR-tANS: non-autoregressive |
> |  | FSAR(O1) | FSAR-tANS: Order-1 Markov |
> |  | FSAR(O2) | FSAR-tANS: Order-2 Markov |
> |  | FSAR(O2)+LSN | FSAR-tANS: Order-2 Markov + LSN |
> |  | FSAR(O3) | FSAR-tANS: Order-3 Markov |

---

> ### Author Response · Authors · 2023-11-23
> **Response to Reviewer QqM2 (Part 3)**
>
> _Q8 - I did not find it in the text and initially was confused why the IN32 results are so much higher than what is typical in the literature._
>
> Regarding the high BPD in IN32, similar findings have been reported in many related studies, such as L3C, bit-swap, and PILC, where it is commonly observed that the BPD in IN32 is typically higher compared to CIFAR10 and IN64.
> On the other hand, as we aimed to target general computation devices, **we opted for a single-stage shallow backbone network to ensure overall computational efficiency**. This choice is discussed in Appendix F.2 on Page 23 of our paper. Specifically:
> > The backbone network comprises a single downsampling (or upsampling) convolutional layer and one residual block within the inference (or generative) model.
>
> For comparison, it is important to note that **L3C employs a much deeper architecture** with a 3-stage 8-resblock-layer network for both inference and generative models. This architecture is superior in terms of likelihood modeling, which explains why their BPD results are typically lower than ours under the same dataset conditions.
> It is worth mentioning that we conducted experiments using a deeper backbone as well, as outlined in Appendix E.4 on Page 21 in the revised paper. For example, the 1-layer backbone with 32 dimension codeword achieve 5.26 BPD while 2-layer backbone with 32 dimension codeword achieve 4.79. These experiments indicated that the use of deeper backbones indeed resulted in lower BPD values.

---

### Official Review · Reviewer_8pa1 · 2023-11-02

**Soundness:** 4 excellent
**Presentation:** 3 good
**Contribution:** 3 good
**Rating:** 6
**Confidence:** 4

**Summary:**

This manuscript proposed a finite-state autoregressive entropy coding method for efficient learned lossless compression, which utilizes a lookup table to expedite autoregressive entropy coding. Specifically, a straight-through hardmax quantization scheme is proposed to enhance the optimization of discrete latent space. Experimental results show that the proposed lossless compression method could improve the compression ratio by up to 6% compared to the baseline, with negligible extra computational time.

**Strengths:**

This paper addresses the shortcomings of existing autoencoder-based codecs and makes targeted improvements from several aspects. The concept of autoregressive modeling based on finite-state Markov model via look-up table is reasonable to alleviate the computational burden of autoregressive models. Besides, an end-to-end adaptive optimization method for selecting learnable state number is proposed to reduce the look-up table size. Furthermore, the straight through hardmax quantization method is proposed for optimizing vector quantized discrete latent space models.

This manuscript is well structured. The technique descriptions is detailed and easy to follow.

Experimental results show that the proposed lossless compression method could improve the compression ratio by up to 6% compared to the baseline, with negligible extra computational time.

**Weaknesses:**

1) Some technique details in Fig. 2 are not clear enough. For example, how to obtain quantize latent y from discrete latent variables z and adaptive state number module?  How to obtain the codebook? There are two bitstream in Fig. 2. Is there any operations proposed for optimizing the data steam?

2) Experimental results are conducted on CIFAR 10 and ImageNet32/64. How about the results on Kodak dataset  and Tecnick dataset? Besides, only the decoding time are provided in Table 1, how about the encoding time?

3) The proposed method achieve RD performance gain and computational efficiency at the cost of memory consumption. Will the size of memory change with different datasets?

**Questions:**

Please refer to the Weakness part.

---

> ### Author Response · Authors · 2023-11-23
> **Response to Reviewer 8pa1 (Part 1)**
>
> We sincerely thank you for your valuable comments and suggestions! We have gone through all your comments and questions, and accordingly conducted additional experiments, provided extra details, and clarified some questions in the revised paper. Below we address each comment or questions individually. The reviewer's comments are shown in italics. The paragraph(s) following them is the authors' response. The quoted paragraphs from our paper are labeled with a grey line left to the paragraphs. Unless specified, all references to pages, equations, and sections refer to the revised paper. Moreover, in the revised version of our paper, we mark all newly added or changed paragraphs in blue. For newly added or changed figures or tables, we mark their titles in blue instead. We will incorporate the suggested revisions into the final camera-ready version to enhance the clarity and persuasiveness of our paper.
>
> _Q1 - Some technique details in Fig. 2 are not clear enough. For example, how to obtain quantize latent y from discrete latent variables z and adaptive state number module? How to obtain the codebook? There are two bitstream in Fig. 2. Is there any operations proposed for optimizing the data steam?_
>
> Reply: We have made updates to Figure 2 to provide more detailed explanations of the vector quantization process, specifically focusing on the red dashed box titled "(b) Straight-through Hardmax Quantization". Moreover, we have added corresponding introductions for the updated figure in Appendix B on Page 14 of the revised paper:
> > To be specific, we calculate the euclidean distances between the inference network output and each codeword in the learnable codebook. The negative euclidean distances are then utilized to determine the sampling probability of each codeword, following a similar approach as in (Sønderby, 2017). Additionally, we incorporate the sparse mask from the LSN module in FSAR to further prune the codebook. The proposed STHQ produces one-hot latent samples by selecting the codeword with the highest probability through quantization. These one-hot samples can be converted into latent indices for compression into the latent stream and are also used in the dequantization process, where a specific codeword is chosen from the codebook as the input for the generative network
>
> It is important to note that we do not impose limitations on the implementation of the data stream coding process. In our compression experiments, we adopt the efficient implementation provided by PILC for data stream coding, as they have already demonstrated its effectiveness.
>
> _Q2 - Besides, only the decoding time are provided in Table 1, how about the encoding time?_
>
> Reply: Thank you for your suggestions! In the revised version of our paper, we have included the encoding time for each method in Table 5 in Appendix E.1 on Page 20:
>
> |  |  | CF10 Comp Time (ms) |  |  | CF10 Decomp Time (ms) |  |  |
> | --- | --- | --- | --- | --- | --- | --- | --- |
> |  |  | Net | Coder | Total | Net | Coder | Total |
> | Continuous | bits-back | 32.07 | 4.54 | 36.61 | 32.44 | 9.85 | 42.29 |
> | Discrete | UQ | 0.98 | 1.53 | 2.51 | 1.60 | 1.54 | 3.15 |
> |  | VQ | 0.98 | 0.61 | 1.59 | 1.57 | 0.88 | 2.45 |
> |  | McQuic | 1.03 | 11.25 | 12.28 | 1.57 | 8.75 | 10.32 |
> | Autoregressive | MaskConv2D | 1.03 | 4.26 | 5.28 | 1.75 | 632.24 | 633.99 |
> |  | Checkerboard | 1.03 | 4.65 | 5.67 | 1.63 | 4.71 | 6.34 |
> | Proposed | tANS | 0.99 | 0.92 | 1.91 | 1.66 | 0.86 | 2.52 |
> |  | FSAR(O1) | 0.98 | 1.00 | 1.98 | 1.58 | 0.97 | 2.55 |
> |  | FSAR(O2) | 0.99 | 1.08 | 2.07 | 1.62 | 1.04 | 2.65 |
> |  | FSAR(O2)+LSN | 0.98 | 1.47 | 2.45 | 1.59 | 1.06 | 2.65 |
> |  | FSAR(O3) | OOM | OOM | OOM | OOM | OOM | OOM |
>
> > The results, displayed in Table 5, indicate that **for most methods, the compression time closely aligns with the decompression time**. However, the MaskConv2D autoregressive method demonstrates a notable distinction, benefitting from its parallel computation during the compression process.
>
> Additionally, we acknowledge that there was an error in the original paper regarding the decompression time reported for the bits-back method. It mistakenly referred to the compression time. However, we would like to assure you that this error does not impact the discussions and conclusions drawn from our results as the compression and decompression time of the bits-back method is close. It has been rectified in the revised version of the paper.

---

> ### Author Response · Authors · 2023-11-23
> **Response to Reviewer 8pa1 (Part 2)**
>
> _Q3 - Experimental results are conducted on CIFAR 10 and ImageNet32/64. How about the results on Kodak dataset and Tecnick dataset?_
>
> Reply: Thank you for your suggestion! In order to evaluate our method's performance on larger scale images, we have followed the approach taken by L3C and PILC. Specifically, we trained a checkpoint using a subset of the Open Images Dataset and conducted lossless compression tests on the Kodak and CLIC20 validation sets. However, However, it is important to note that due to limitations in terms of time and computational resources, we were only able to train on a smaller subset of approximately 100k images. This subset is smaller than that used by PILC, which employed around 400k images according to the L3C official code. Consequently, the resulting BPD values may be worse than usual.
>
> The results of these experiments can be found in Table 6, which is included in Appendix E.2 on Page 20 of the revised version of our paper. To provide further clarity, we would like to quote the relevant information from Appendix E.2 as follows:
>
> |  |  | Practical BPD |  |  |  |  | Speed (MB/s) |  |
> | --- | --- | --- | --- | --- | --- | --- | --- | --- |
> |  |  | CF10 | IN32 | IN64 | Kodak | CLIC20 | CSpd | DSpd |
> | Traditional | PNG | 5.78  | 6.09  | 5.42  | 4.53  | 4.23  | 16.559  | 59.207  |
> |  | WebP | 4.52  | 4.95  | 4.32  | 3.20  | 3.08  | 1.762  | 26.688  |
> |  | FLIF | 4.29  | 4.78  | 4.25  | 2.88  | 2.82  | 0.858  | 2.715  |
> | Autoencoder | L3C | 4.77  | 4.89  | 4.29  | 3.55  | 3.34  | 0.036  | 0.057  |
> |  | bitswap | 3.84  | 4.54  | - | - | - | 0.003  | 0.004  |
> |  | PILC | 4.28  | 4.95  | 4.51  | 3.86  | 3.79  | 2.659  | 3.725  |
> | Flow | IDF | 3.34 | 4.18  | 3.90  | - | - | 0.010  | 0.007  |
> |  | IDF++ | 3.26  | 4.12  | 3.81  | - | - | 0.009  | 0.007  |
> | Proposed | PILC+STHQ | 4.29  | 4.90  | 4.50  | 3.80  | 3.72  | 2.566  | 3.795  |
> |  | PILC+STHQ+FSAR | 4.18  | 4.80  | 4.39  | 3.74  | 3.65  | 2.378  | 3.590  |
>
> > We present comprehensive results in Figure 4, encompassing experiments conducted on ImageNet32/64, as well as a high-resolution benchmark utilizing the Kodak  (Kod, 2023) dataset and CLIC 2020 (CLI, 2023) validation set (referred as to Kodak and CLIC20 respectively).  The Kodak dataset consists of images with a resolution of $512\times 768$, while the CLIC dataset ranges from $720\times 439$ to $2048\times 1370$ in resolution. It is important to note that our model is trained following the methods described in (Mentzer et al., 2018) and (Kang et al., 2022), using a subset of the preprocessed Open Images training dataset (Kuznetsova et al., 2018) consisting of approximately 90,000 images. Furthermore, our evaluation includes lossless codecs based on flow models, specifically integer discrete flows (IDF) (Hoogeboom et al., 2019) and its enhanced version IDF++ (van den Berg et al., 2020). The corresponding results are presented in Table 6.
>
> > The results of the PILC+STHQ+FSAR method on IN32/64 demonstrate similarity to CIFAR10, exhibiting similar BPD values to WebP and FLIF, and slightly inferior performance compared to PILC. **In the case of high-resolution images, the PILC+STHQ+FSAR method performs marginally worse than WebP but outperforms PNG, likely due to the limited versatility of the shallow backbone. This implies the necessity for improved backbones when dealing with large-scale images. However, it is important to note that the proposed STHQ and FSAR techniques enhance the performance of PILC across all datasets, validating the effectiveness and robustness of the proposed method for lossless image compression tasks. Regarding flow models, IDF and IDF++ surpass all other methods in terms of BPD, although their slower processing speed remains a critical concern**.

---

> ### Author Response · Authors · 2023-11-23
> **Response to Reviewer 8pa1 (Part 3)**
>
> _Q4 - The proposed method achieve RD performance gain and computational efficiency at the cost of memory consumption. Will the size of memory change with different datasets?_
>
> Reply: The memory consumption of the proposed method is primarily determined by the order of the Markov model and the number of states (codewords) in the FSAR-based entropy coder, as explained in the "Learnable State Number" paragraph in Section 3, Page 5:
> > **Learnable State Number** We notice that the size of the lookup table $\mathcal{O}(C^{N})$ can become prohibitively large, especially when $N \ge 2$. A common practice to optimize C is through hyperparameter tuning...
>
> **Therefore, the memory size will not change with different dataset unless we use a codebook with different states or order for each dataset.**
>
> Furthermore, we have also discussed how to potentially further reduce memory usage in the original paper in Appendix G "Discussion and Future Works" as follows:
>
> > Despite the promising outcomes achieved through the proposed finite-state autoregressive entropy coding approach, there are still certain limitations that require attention. Notably, our experimental utilization of the lookup table still demands approximately 300MB of storage, even after reducing the number of states using the proposed Learnable State Number approach. This presents a significant hurdle for implementing the approach on portable devices and emphasizes the necessity for future research on maximizing the advantages of the autoregressive Markov model while keeping the space complexity limited.
>
> > One potential solution is to restrict the number of probability distributions employed for tANS coding, thereby limiting the corresponding state transition lookup tables used by tANS. By doing so, we only need to store pointers to these restricted distributions in the FSAR lookup table L. Assuming we limit the distributions to $D$ and each corresponding state transition lookup table contains $S$ states, for order-2 FSAR models, the minimal memory requirement is $ DS \lceil \log_{256}S \rceil + C^{2} \lceil \log_{256}D \rceil$ bytes. In contrast, unlimited distributions would necessitate $ C^{2} S \lceil \log_{256}S \rceil $ bytes. In our experiments, setting $D=256$ reduced memory usage from approximately 512MB to around 2MB. We leave determining the optimal selection of these limited distributions to achieve minimal performance degradation to future work.

---

### Meta-Review · Area_Chair_sVaZ · 2023-12-10

**Metareview:**

This paper proposes a finite-state autoregressive entropy coding for lossless compression. Reviewers agree that this is a well-written paper with solid technical contributions. The proposed method can improve the compression ratio without sacrificing efficiency. The proposed STHQ method is interesting. Weaknesses include higher memory consumption than previous methods.

**Justification For Why Not Higher Score:**

This paper has higher memory consumption than previous methods.

**Justification For Why Not Lower Score:**

Overall this is a technically solid paper and authors did well on addressing reviewers' concerns and improving their paper.

---

### Decision · Program_Chairs · 2024-01-16

Accept (spotlight)